# A stochastic world model on gravity for stability inference

**Taicheng Huang, Jia Liu***

Department of Psychological and Cognitive Sciences & Tsinghua Laboratory of Brain and Intelligence, Tsinghua University, Beijing, China

**Abstract** The fact that objects without proper support will fall to the ground is not only a natural phenomenon, but also common sense in mind. Previous studies suggest that humans may infer objects' stability through a world model that performs mental simulations with a priori knowledge of gravity acting upon the objects. Here we measured participants' sensitivity to gravity to investigate how the world model works. We found that the world model on gravity was not a faithful replica of the physical laws, but instead encoded gravity's vertical direction as a Gaussian distribution. The world model with this stochastic feature fit nicely with participants' subjective sense of objects' stability and explained the illusion that taller objects are perceived as more likely to fall. Furthermore, a computational model with reinforcement learning revealed that the stochastic characteristic likely originated from experience-dependent comparisons between predictions formed by internal simulations and the realities observed in the external world, which illustrated the ecological advantage of stochastic representation in balancing accuracy and speed for efficient stability inference. The stochastic world model on gravity provides an example of how a priori knowledge of the physical world is implemented in mind that helps humans operate flexibly in open-ended environments.

***For correspondence:**
liujiathu@tsinghua.edu.cn

**Competing interest:** The authors declare that no competing interests exist.

## eLife assessment

In this **valuable** study, the authors present findings that suggest that people do not faithfully replicate the physics of the real world but rather have a stochastic world model, specifically a stochastic representation of gravity. This contrasts with prior accounts that suggested a potentially noisy Newtonian model where the noise arises from perceptual uncertainty or (inferred) external perturbations. The experimental evidence is generally **solid**, with all experiments and model simulations being consistent with the proposed account. In the revision, the authors also added a number of control experiments that address some of the most pressing concerns of the original submission.

## Introduction

About 2000 years ago, Confucius warned his disciples that a wise man should not stand next to a collapsing wall. We, wise or not, can easily judge whether a wall is stable or collapsing in a fraction of a second (*Battaglia et al., 2013*; *Kubricht et al., 2017*; *McCloskey, 1983*). This astonishing performance is unlikely to have been achieved by previous visual experience alone. Taking a stack consisting of 10 blocks as an example (*Figure 1*), we can quickly report its stability with a satisfactory accuracy of 70% on average (*Bear et al., 2021*; *Zhang et al., 2016*), but the universal cardinality of possible configurations is at least $1.14 \times 10^{50}$ (*Figure 1—figure supplement 1*), which is much larger than the total number of sand grains on Earth (est. $7.5 \times 10^{18}$) (*Blatner, 2013*). Contrary to this intuition, 4-month-old infants, who have little visual experience of the physical world, expect a box to fall if it loses contact with a support platform (*Baillargeon, 2004*; *Baillargeon, 1994*). Our minds may therefore have devised a mechanism that differs from the widely used discriminative approach in artificial

neural networks, which relies on the extensive visual experience of objects and feedback about their stability (*Bear et al., 2021*; *Li et al., 2016*; *Zhang et al., 2016*).

Indeed, both behavioral and neuroimaging studies have suggested that humans possess a priori knowledge of Newton's law of physics in the mind. For example, infants as young as 7 months expect a downward-moving object to accelerate and an upward-moving object to decelerate (*Friedman, 2002*; *Kim and Spelke, 1999*), and adults can estimate the remaining time to catch a moving ball (*McIntyre et al., 2001*; *Zago and Lacquaniti, 2005*) even in the absence of visual information (*Lacquaniti and Maioli, 1989*; *Zago et al., 2009*). Further fMRI studies have revealed the parieto-insular vestibular cortex in the brain as the neural basis for gravity-based stability inference, suggesting that this knowledge is encapsulated as a cognitive module (*Fischer et al., 2016*; *Indovina et al., 2005*; *Pramod et al., 2022*). Accordingly, our brain is proposed as a set of generative machines that actively predict future events of the ever-changing physical world through mental simulation with a priori knowledge acting upon the world (*Battaglia et al., 2013*; *Hegarty, 2004*; *Huang and Rao, 2011*; *Tenenbaum et al., 2011*; *Ullman et al., 2017*). For this reason, the generative machine is also called the world model (*Land, 2014*; *Tenenbaum et al., 2011*).

Recently, the idea of the world model has become popular to explain the predictive nature of the brain (*Friston et al., 2021*) and improve the generality and robustness of the artificial neural networks (*Matsuo et al., 2022*). However, how a priori knowledge is implemented in the world model remains to be determined. A prevailing theory suggests that the world model in the brain accurately mirrors the physical laws of the world (*Allen et al., 2020*; *Battaglia et al., 2013*; *Zhou et al., 2022*). For example, the direction of gravity encoded in the world model, a critical factor in stability inference, is assumed to be straight downward, aligning with its manifestation in the physical world. To explain the phenomenon that tall and thin objects are subjectively perceived as more unstable compared to short and fat ones (*Figure 1—figure supplement 2*), external noise, such as imperfect perception and assumed external forces, is introduced to influence the output of the model. However, when the brain actively transforms sensory data into cognitive understanding, these data can become distorted (*Kriegeskorte and Douglas, 2019*; *Naselaris et al., 2011*), hereby introducing uncertainty into the representation of gravity's direction. In this scenario, the world model inherently incorporates uncertainty, eliminating the need for additional external noise to explain the inconsistency between subjective perceptions of stability and the actual stability of objects. Note that this distinction of these two theories is nontrivial: the former model implies a deterministic representation of the external world, while the latter suggests a stochastic approach. Here, we investigated these two alternative hypotheses regarding the construction of the world model in the brain by examining how gravity's direction is represented in the world model when participants judged object stability. Here, we investigated these two alternative hypotheses for the construction of the world model in the brain by examining how gravity's direction was represented in the world model when participants judged the stability of objects.

To do this, we measured participants' sensitivity to gravity's direction in a stability inference task (*Battaglia et al., 2013*) and found that gravity's direction was encoded in a Gaussian distribution, with the vertical direction as the maximum likelihood. This stochastic parameter was then built into the world model to simulate the displacement of blocks in a stack under the force of gravity, and the simulation result fits nicely with participants' judgment of stacks' stability and explains the daily illusion that taller objects are perceived as more likely to fall. A computational model with a reinforcement learning (RL) algorithm was devised to reveal its origin through interactions with the physical world. Finally, we explored the ecological advantage of the stochastic feature of the world model.

## Results

### The direction of gravity in the world model

The direction of gravity is perpendicular to the ground surface. Here, we first tested humans' sensitivity to gravity's direction to investigate how faithfully our gravity is represented in the world model compared to gravity in the physical world. To do this, we used Pybullet (*Coumans and Bai, 2016*), a forward physics simulator, to manipulate gravity's direction. Then, we asked the participants to judge whether the collapse trajectories of unstable stacks were normal (*Figure 1a*, *Figure 1—video 1*). The direction of simulated gravity was measured by a parameter pair $(\theta, \varphi)$ (*Figure 1b*), which determines

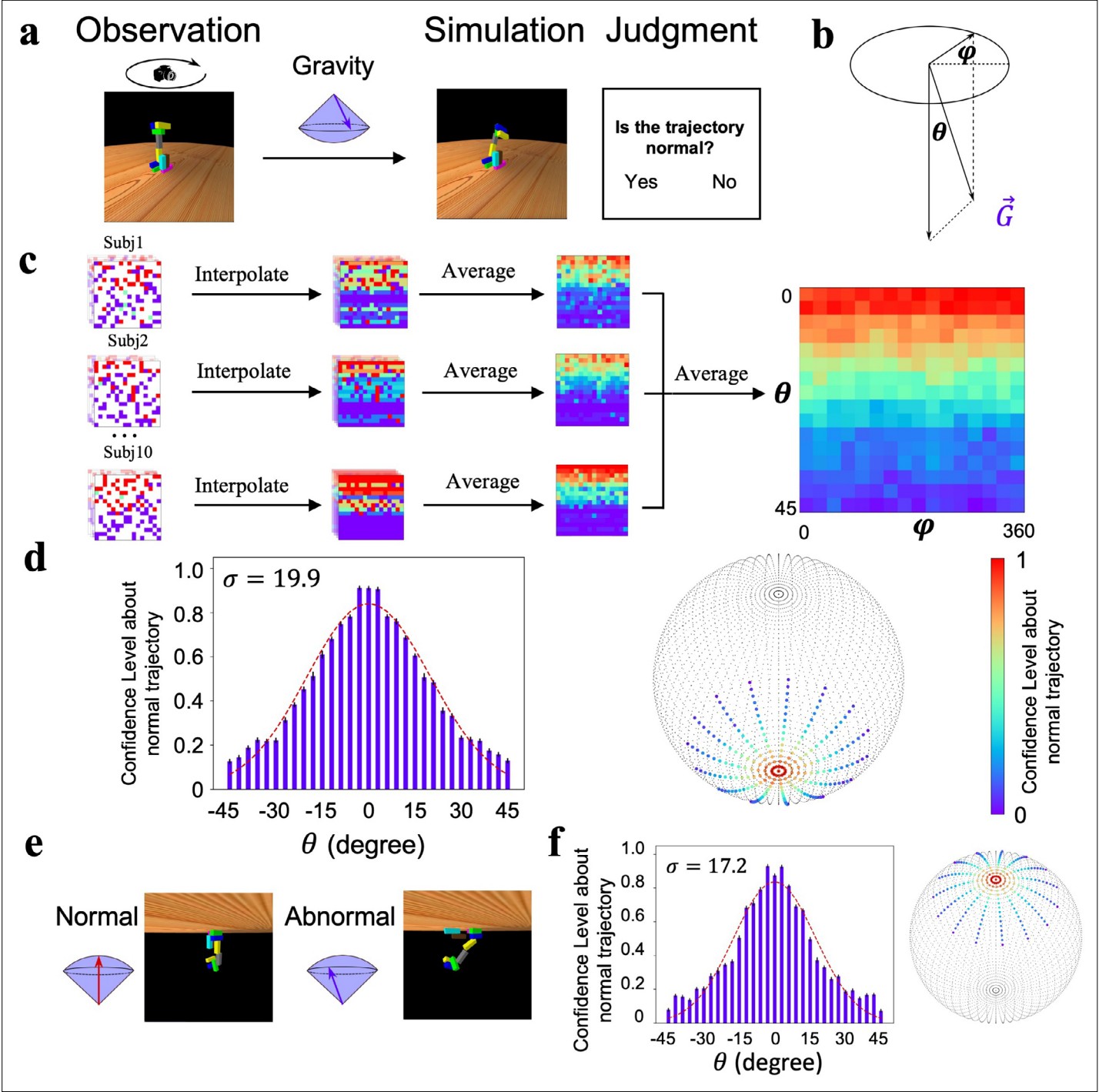

**Figure 1.** Gravity's direction in the world model. (**a**) The design of the behavioral experiment. Left: a rotating camera was used to rotate a stack 360° to display the three-dimensional appearance of the configuration. Middle: gravity's direction was randomly sampled from a spherical surface. Right: the physics simulator simulated the collapse trajectory of the stack under this selected direction, and participants reported whether the collapse trajectory was normal. (**b**) The spherical surface of gravity's direction was determined by two parameters $\theta$ and $\varphi$. (**c**) The procedure of calculating participants' confidence level about normal trajectory as the function of angle pairs. Left: each cell represents the response of normal trajectory for an angle pair within a run. Middle: responses for unsampled pairs were interpolated with the averaged responses along $\varphi$. Right: the confidence level for each angle pair was calculated by averaging responses across runs and participants. (**d**) Left: gravity's direction encoded in the world model follows a Gaussian distribution with the vertical direction as the maximum likelihood. Note that the confidence level for $\theta > 0$ was sampled from $\varphi \, \epsilon \, (0°, 180°)$, and for $\theta < 0$ was sampled from $\varphi \, \epsilon \, (180°, 360°)$. Right: the sphere represents the space of gravity's direction, with two poles pointing upward and downward, respectively. Each dot in the sphere represents one angle pair, and the color on a dot indicates the likelihood that the collapse trajectory under this

*Figure 1 continued on next page*

*Figure 1 continued*

gravity direction was judged normal. (**e**) In a new setting, gravity's direction is reversed. Left: an example collapse trajectory when gravity's direction was upward. Right: a trajectory when the direction was away from the vertical upward. (**f**) Gravity's direction encoded in the world model when gravity's direction in the physical world was reversed. Error bar: standard error, sample size=10.

The online version of this article includes the following video and figure supplement(s) for figure 1:

**Figure supplement 1.** Construction of stacks with different configurations.

**Figure supplement 2.** Differentiating subjectivity from objectivity.

**Figure supplement 3.** The stochastic world model on gravity of each participant.

**Figure supplement 4.** Wall experiment to test the impact of external forces on the measurement of stochastic gravity.

**Figure supplement 5.** The stochastic world model on gravity of each participant when gravity's direction was inverted.

**Figure 1—video 1.** Collapse simulation under normal or abnormal gravity in an upright perspective.

https://elifesciences.org/articles/88953/figures#fig1video1

**Figure 1—video 2.** Collapse simulation under normal or abnormal gravity in an upright perspective.

https://elifesciences.org/articles/88953/figures#fig1video2

the deviation of the direction of simulated gravity from the direction of gravity in the physical world. Specifically, $\theta$ is the vertical component of the direction that affects the degree of collapse, and $\varphi$ is the horizontal component that determines the orientation of collapse. We collected participants' judgment of the normality of collapse trajectories while varying $\theta$ from 0° to 45° and $\varphi$ from 0° to 360° across the force space, and the confidence level of the judgment for each angle pair was used to index participants' sensitivity to gravity's direction (*Figure 1c*). As expected, when $\theta$ is equal to 0 (i.e., the direction of the simulated gravity is the direction of the natural gravity), the participants were likely to report that the collapse trajectory was normal (accuracy: 91.0%, STD: 8.0%). Then, the critical question is how participants' subjective sense of the normal degree of collapse trajectories changes as a function of $\theta$. If our world model on gravity is a faithful replica of the physical reality, we should expect the immediate detection of abnormality when $\theta$ is away from 0.

Contrary to this intuition, the subjective sense of the abnormality was not immediately apparent as $\theta$ moved away from 0; instead, the confidence level of reporting normal trajectories decreased gradually as a function of $\theta$, which was the best fit by a Gaussian function with $\sigma = 19.9$ (*Figure 1d*, left). That is, the participants were 50.9% confident in reporting a normal collapse trajectory when the vertical offset of $\theta$ was 19.9°. In addition, accuracy in detecting the abnormality was not affected by $\varphi$ (*Figure 1—figure supplement 3*), consistent with the uniformly distributed gravitational field in the physical world. This pattern was observed for all participants tested, with $\sigma$ varying from 11.1 to 37.1 (*Figure 1—figure supplement 3*), and remained unchanged with the addition of a wall on one side to block potential external disturbances from wind (*Figure 1—figure supplement 4*). Therefore, the world model on gravity is unlikely to be a faithful replica of the physical world; instead, it encodes gravity's direction as a Gaussian distribution with the vertical direction as the maximum likelihood (*Figure 1d*, right).

To further test whether the world model on gravity, once established, is encapsulated from visual experience and task context, we inverted the virtual environment upside down with gravity's direction pointing upward, and then asked the same group of participants to judge whether collapse trajectories were normal (*Figure 1e*, *Figure 1—video 2*). We found that the confidence level also decreased gradually as a function of $\theta$ (*Figure 1f*, $\sigma = 17.2$; see *Figure 1—figure supplement 5* for each participant), which was not significantly different from that in the environment with gravity pointing downward. Indeed, each participant's $\sigma$ in the upright condition was in high agreement with the $\sigma$ in the upside-down condition ($r = 0.91$, p<0.01). That is, the visual experience and task context apparently did not cognitively penetrate humans' world model on gravity, suggesting that it is likely encapsulated as a cognitive module.

How does the stochastic gravity's direction in the world model affect our inference on objects' stability? To answer this question, we recruited an independent group of participants to estimate the stability of 60 stacks of different configurations (*Figure 2a*), half of which were stable. During the experiment, the participants were required to judge how stable each stack was on a 0–7 scale without feedback, which was used to index their subjective sense about stacks' stability. Two world models

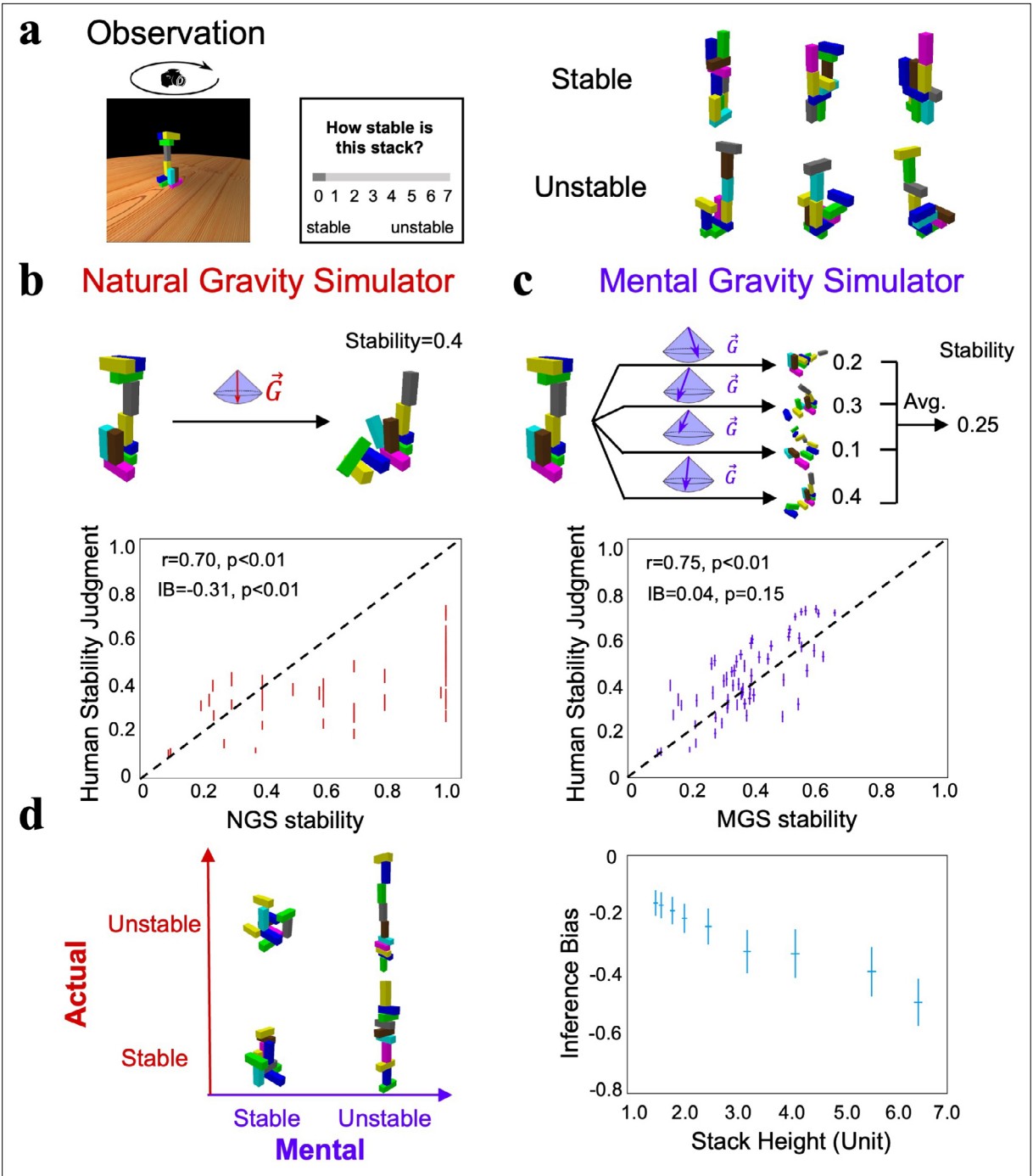

**Figure 2.** Stability inference by the world model on gravity. (**a**) An experiment to rate the stability of stacks, half of which were stable and the other half unstable. (**b**) Top: the procedure of the natural gravity simulator (NGS) to estimate the actual stability of stacks by simulation, and for unstable stacks the stability was indexed by the proportion of displaced blocks. Bottom: the correlation between the stability estimates of the participant and those of the NGS. Each dot represents one stack, and the lines denote the standard errors. (**c**) Top: the procedure of the mental gravity simulator (MGS), where the stability of a stack was estimated by averaging the estimated stabilities from multiple simulations with different gravity directions sampled from the Gaussian distribution. Bottom: the correlation between the stability estimates of the participant and those of the MGS. (**d**) Left: the illusion that taller objects are perceived as more unstable than shorter ones. Right: the inference bias was indexed by the difference between the stability estimated by the MGS and that estimated by the NGS. The larger the negative values, the more likely stacks were unstable. The x-axis denotes the height of a stack containing ten blocks, where the height, length, and width of each block were 1.2, 0.4, and 0.4, respectively. IB: inference bias. Error bar: standard error.

The online version of this article includes the following figure supplement(s) for figure 2:

*Figure 2 continued on next page*

*Figure 2 continued*

**Figure supplement 1.** Relation between the stability estimated by the mental gravity simulator (MGS) stability and that by participants when the world model was implemented with different Gaussian functions.

**Figure supplement 2.** Height illusion of stability inference when the world model was implemented with different Gaussian functions.

were constructed for comparison. One world model was equipped with a vertically downward direction of gravity without any stochastic variance. This deterministic model is intended to simulate how the stacks fell in the real world, and is therefore called a natural gravity simulator (NGS) (*Figure 2b*, top). The other model is the same as the NGS, except that the deterministic direction of gravity in the NGS was replaced by the stochastic direction obtained from the previous psychophysical experiment. This model is thus called the mental gravity simulator (MGS, *Figure 2c*, top). Both models were used to quantify the degree of stability by measuring the proportion of unmoved blocks after the collapse, where the proportion of unmoved blocks after the simulation was used to estimate the stability of the stacks.

NGS-estimated stability was significantly correlated with participants' subjective sense (*Figure 2b*, bottom; $r = 0.70$, $p<0.01$), consistent with previous findings (*Battaglia et al., 2013*). However, the participants showed an obvious bias towards predicting a collapse for stacks regardless of their actual stability, as the dots in *Figure 2b* are more concentrated on the lower side of the diagonal line. This phenomenon is referred to as the inference bias, which was indexed as the difference in stability estimates between the participants and the NGS (inference bias = $-0.31$, $p<0.01$; see 'Methods'). In other words, the participants were unlikely to infer stacks' stability from simulations with a deterministic direction of gravity pointing vertically downward. In contrast, the MGS randomly sampled pairs of $(\theta_s, \varphi_s)$ from the Gaussian distribution as gravity's directions 100 times, and the estimated stability of a stack was the averaged stability of simulations with different angle pairs. Aside from a similar magnitude of the correlation in the stability estimates between the participants and the MGS (*Figure 2c*, bottom; $r = 0.75$, $p<0.01$), the MGS, in contrast to the NGS, more precisely reflected participants' judgments of stability because the points were evenly distributed along the diagonal line (inference bias = $0.04$, $p>0.05$; see *Figure 2—figure supplement 1* for the agreement when the MGS was implemented with different Gaussian functions). In other words, the magnitude of the correlation coefficients is not the only indicator to evaluate the model's fitness. In short, the world model that represents gravity's direction as a Gaussian distribution around the vertical direction properly explains our tendency to judge stacks as more prone to collapse.

The stochastic world model illustrated by the MGS that led to participants' inference bias may explain the daily illusion that we perceive taller objects to be more unstable than shorter ones (*Figure 2d*, left). An intuitive explanation from physics is that a tall object has a higher center of gravity, and thus an external perturbation makes it more likely to collapse. Our stochastic world model, on the other hand, provides an alternative explanation without introducing external perturbations because the center of gravity in taller objects is more susceptible to influence when gravity deviates slightly from a strictly downward direction during humans' internal simulations. To test this conjecture, we constructed a set of stacks with different heights and estimated the degree of stacks' stability with the MGS and the NGS, respectively. Because the MGS was considered to be the world model implemented in the brain, the inference bias here was calculated as the difference in stability estimates between the MGS and the NGS, with negative values indicating a tendency to judge a stable stack as an unstable one. Consistent with the inference bias found in humans, the MGS found stacks of all heights to be more prone to collapse (*Figure 2d*, right; inference bias <0, $p<0.01$ for all heights). Critically, the bias increased monotonically with increasing height, consistent with the illusion that taller objects are considered more prone to collapse (see *Figure 2—figure supplement 2* for the inference bias when the MGS was equipped with different levels of deviation). In short, the stochastic world model on gravity provides a more concise explanation for the daily illusion that taller objects are perceived as more likely to collapse, without assuming external perturbations.

## The origin of the stochastic feature of the world model

A deterministic model that combines gravity's veridical direction with external perturbations, such as an external force or perceptual uncertainty (*Allen et al., 2020*; *Battaglia et al., 2013*; *Lake et al., 2017*; *Smith and Vul, 2013*), is theoretically equivalent to our stochastic model that represents gravity's

direction in a Gaussian distribution; therefore, it also fits well with humans' inference on stability by fine-tuning the parameters of external perturbations. While the cognitive impenetrability and the self-consistency observed in this study, without resorting to an external perturbation, favor the stochastic model over the deterministic one, the origin of this stochastic feature of the world model is unclear.

Here we used an RL framework to unveil this origin because our intelligence emerges and evolves under the constraints of the physical world. Therefore, the stochastic feature may emerge as a biological agent interacts with the environment, where the mismatches between external feedback from the environment and internal expectations from the world model are in turn used to fine-tune the world model (*Friston et al., 2021*; *MacKay, 1956*; *Matsuo et al., 2022*). Note that a key aspect of the framework is determining whether the stochastic nature of the world model on gravity emerges through this interaction, even in the absence of external noise. To simulate this process, here we designed an RL framework to model this interactive process to illustrate how the world model on gravity evolves (*Figure 3a*). Specifically, an agent perceived a stack in the environment, which was then acted upon by a simulated gravity with direction parameters (i.e., $\theta$ and $\varphi$) sampled from a spherical direction space. The initial probabilities for the sampling directions were identical (*Figure 3b*, left). The final state of the stack served as the agent's expectation under the effect of the simulated gravity. The mismatch between the expectation and the observed final state of the stack under the natural gravity was used to update the sampling probability of the direction space, with a larger discrepancy leading to a larger decrease in probabilities through RL. Within this RL framework, we constructed abundant stacks of 2–15 blocks to train the world model on gravity. As the training progressed, the probabilities of the direction space gradually converged downward (*Figure 3b*, middle; see *Figure 3—figure supplement 1* for the training trajectory). Although gravity's direction in the environment was vertical, the distribution of updated probabilities in the direction space was gradational ($\sigma$ = 21.6; *Figure 3b*, right), which is close to gravity's direction represented in the world model derived from the psychophysics experiment on human participants. Therefore, the world model representing gravity's direction in a Gaussian distribution can emerge automatically as the agent interacts with the environment, without the need for any external perturbation.

To further illustrate the idea that the environment constrains the form of intelligence, we systematically manipulated the appearance of the physical world while holding the natural gravity constant. Specifically, we constructed 14 worlds, each containing stacks of the same number of blocks, but with different configurations. The number of blocks ranged from 2 to 15. We trained the world model on gravity under the same RL framework for each world and found that all world models represented gravity's direction in a Gaussian distribution (*Figure 3c*, left; see *Figure 3—figure supplement 2* for all world models). However, the width of the distribution, indexed by the parameter of $\sigma$, decreased monotonically as the number of blocks increased (*Figure 3c*, right). This phenomenon was shown because in general stacks containing more blocks were more likely to be affected by forces whose directions were not perpendicular to the ground surface, which provided more information about gravity, and thus resulted in a more accurate representation of gravity's direction in the world model. In short, the world model on gravity resonates with not only the physical law governing the environment, but also the specific regularities of the environment the agent encountered.

## The ecological advantage of the stochastic world model

When passing a cliff face, we have to be constantly aware of the stability of the rocks on the cliff. The ideal response would be both accurate and fast, but accuracy and speed are often difficult to achieve simultaneously. Here we investigated how the world model on gravity balances these two factors with its stochastic feature. To answer this question, we used a linear classifier (i.e., logistic regression) to model humans' decision-making behavior at different stages of the mental simulation. Specifically, we collected all the position coordinates of a stack's blocks at different stages of the simulation. The position difference between the intermediate states of the stack and the initial state provides information about the stability of the stack. For example, a stable stack should have no difference in the positions of the component blocks at all simulation stages, and an unstable stack should have a gradually increasing position difference. If the linear classifier detected the difference in positions sufficient for the classification at any stage, it classified the stack as unstable, otherwise stable (*Figure 4a*). The classification accuracy gradually increased as the simulation progressed until it reached the asymptote.

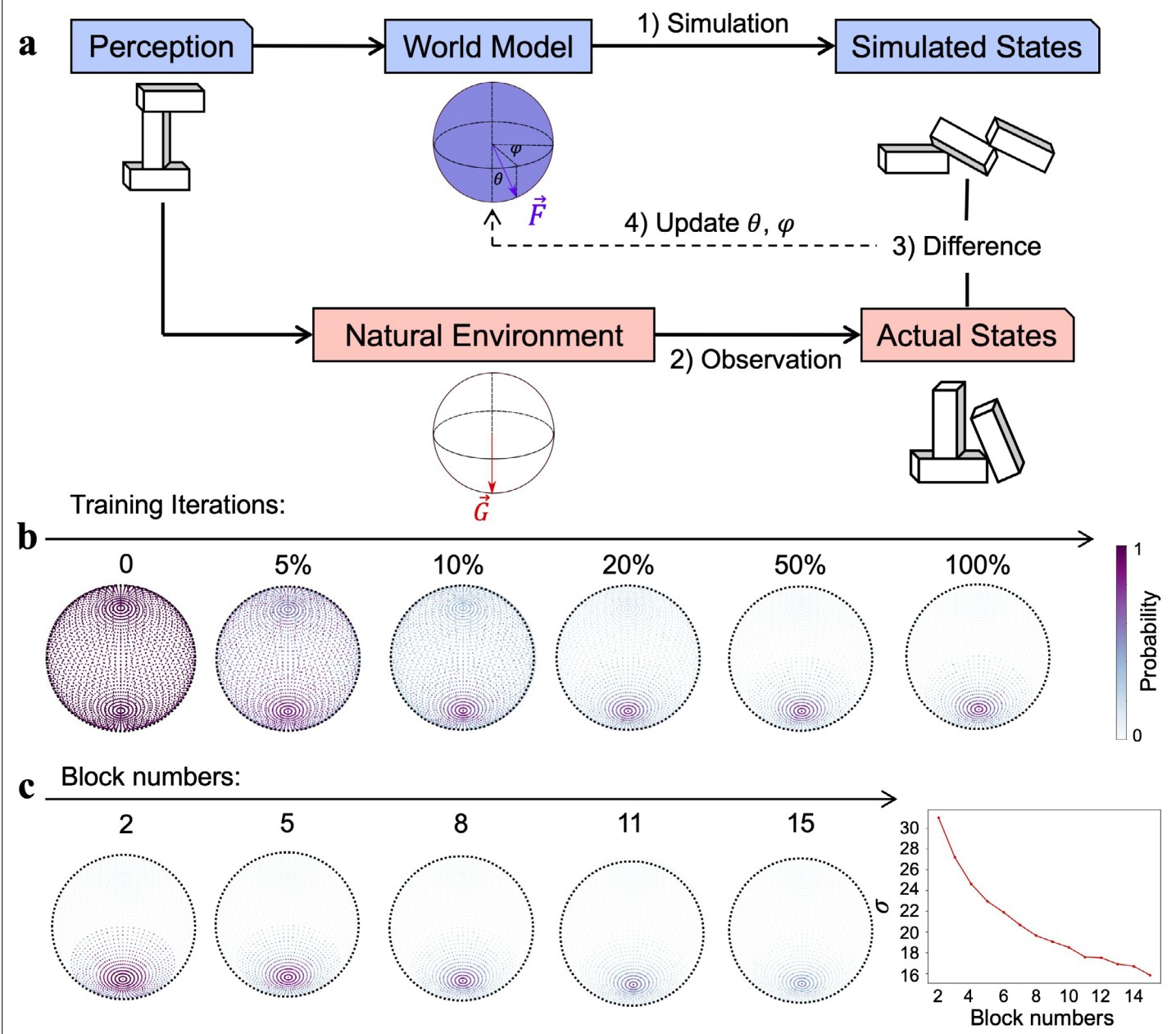

**Figure 3.** The origin of the stochastic feature of gravity's direction. (**a**) The reinforcement learning framework, which updated gravity's direction $(\theta, \varphi)$ of the world model by minimizing the difference between the expectation from the internal simulation (i.e., simulated states) and the observation from the physical world (i.e., actual states). (**b**) Gravity's directions, which were uniformly distributed on the spherical surface, gradually converged downward as the training progressed and eventually stabilized in a Gaussian distribution with the vertical direction as the maximum likelihood. Color denotes the probability of a parameter pair being adopted as gravity's direction. (**c**) Left: world models constructed by reinforcement learning when stacks in the physical world were composed of different numbers of blocks ranging from 2 to 15. Right: the variance of the Gaussian distribution, illustrated by the width of the distribution of gravity's direction on a spherical surface, monotonically decreased as the number of blocks in the stacks increased.

The online version of this article includes the following figure supplement(s) for figure 3:

**Figure supplement 1.** The developmental trajectory of $\theta$ (top) and $\varphi$ (bottom) angles.

**Figure supplement 2.** The world models developed in the world containing stacks with different numbers of blocks.

As expected, for the NGS (i.e., the world model with the deterministic direction of gravity), the accuracy at the plateau was close to 100% (95.3% on average, *Figure 4b*, top red box), significantly higher than that for the MGS (80.1% on average, *Figure 4b*, top blue box) ($t = 19.59$, $p<0.001$), simply because of the stochastic feature of gravity's direction. However, while the initial growth rates of both

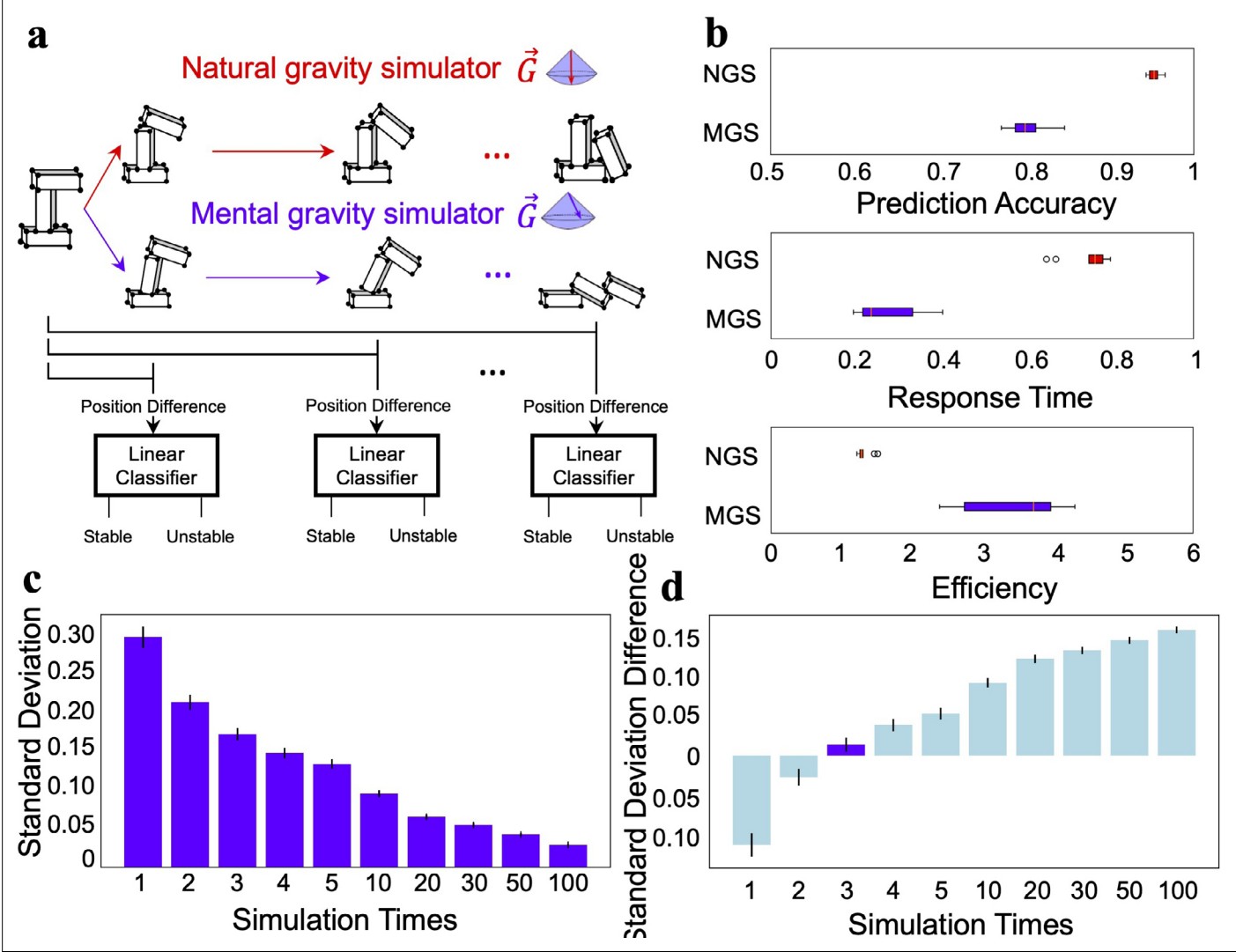

**Figure 4.** The ecological advantage of the stochastic feature. (**a**) Illustration that modeled humans' decision-making behavior at different stages of the mental simulation using the natural gravity simulator (NGS) and mental gravity simulator (MGS). (**b**) The decision of the linear classifier based on the simulation of the MGS was less accurate than that of the NGS (top), but the decision was made faster in the MGS than in the NGS (middle). The MGS was more efficient than the NGS in combining accuracy and speed (bottom). (**c**) The relationship between the number of simulations and the variance of the estimated stability. (**d**) The difference in the variance of the estimated stability between the participants and the MGS. The difference was minimal when the MGS ran the simulation three times. Error bar: standard error.

The online version of this article includes the following figure supplement(s) for figure 4:

**Figure supplement 1.** Ecological advantage of the world model embodied with different Gaussian functions.

**Figure supplement 2.** The relation between the number of simulations and the variance of stability inference.

models were comparable, the MGS reached the plateau crucial for decision-making sooner than the NGS (response time, indexed by the ratio between the time to reach the plateau and the time to reach the final stage: 27.1% vs 75.2%, $t = 15.58$, p<0.001) (*Figure 4b* middle). The same pattern was also observed with different variances of the Gaussian distribution (*Figure 4—figure supplement 1*). That is, the stochastic world model prioritized speed over accuracy, echoing the basic principle of survival: fleeing potential danger as quickly as possible, rather than making a perfect decision with a dreadful delay. In addition, by integrating the prediction accuracy and the response time as a measure of efficiency, we found that the stochastic world model provided a better balance between accuracy and speed, with an efficiency significantly higher than that provided by the NGS (3.49 vs 1.32, $t = 9.12$, p<0.001; *Figure 4b*, bottom).

On the other hand, if time permits, multiple simulations with the MGS can significantly reduce the variance introduced by the stochastic representation of gravity's direction (*Figure 4c*). To explore whether humans adopted this strategy of performing multiple simulations before making a decision, we ran simulations with the MGS at different numbers of times and then matched them with humans' performance. We found that the variance of humans' inference on stability best matched that of the MGS after three simulations (*Figure 4d*; see *Figure 4—figure supplement 2* for the model-behavior correspondence under different numbers of simulations). Therefore, humans are likely to run simulations a limited number of times to infer stacks' stability.

## Discussion

In this study, we investigated how the physical law of gravity is embodied in the brain as a world model that guides inferences on objects' stability. A series of psychophysics experiments showed that the world model on gravity is not a faithful replica of the physical world, but rather a stochastic model that captures the essence of the vertically downward direction of gravity as the maximum likelihood of a Gaussian distribution. The stochastic feature of the world model not only fits humans' stability inference behavior better than the deterministic model, but also provides new insight into the daily illusion that taller objects are perceived as more likely to collapse. We further illustrated how the stochastic feature evolved through interactions with the environment using RL, and well-balanced accuracy and speed to produce a unique ecological advantage for our survival in the physical world.

About 300 years ago, the philosopher Immanuel Kant proposed the intuition of space and time as a priori knowledge in the mind for us to understand the physical world (*Kant, 1781*), but only until recently have researchers investigated how the intuition is implemented in the brain as intuitive physics (*Kubricht et al., 2017*; *McCloskey, 1983*). In the Noisy Newtonian Framework, intuitive physics is depicted as a combination of Newtonian physics and uncertainty generated by noise (*Battaglia et al., 2013*; *Kubricht et al., 2017*; *Sanborn et al., 2013*). The introduction of uncertainty helps to reconcile the misconception occurring under unfavorable conditions, such as unfamiliar events or static scenes (*Kaiser et al., 1992*; *Kaiser et al., 1986*; *Kim and Spelke, 1999*; *McCloskey, 1983*; *Smith and Vul, 2013*), which was once thought to support Aristotelian physics (*Disessa, 1982*; *Halloun and Hestenes, 1985*). The noise in previous studies was thought to originate from sources such as perceptual uncertainty or external perturbations of forces, rather than from the intuitive physical engine itself, which is thought to be a deterministic system. Our study extends these deterministic models by showing a stochastic world model that the noise instead came from the representation of gravity's direction under Gaussian distribution. The inherent stochastic feature of gravity's direction did not need to rely on external noise to explain the illusory instability of taller objects. In addition, it was also confirmed by the cognitive impenetrability of the Gaussian distribution of gravity's direction when gravity's direction in the physical world was reversed (*Pylyshyn, 1980*).

With an RL framework, we further proposed a possible origin of the stochastic feature of the world model through interactions with the physical world. In contrast to summarizing statistical patterns from experience (*Bear et al., 2021*; *Li et al., 2016*; *Zhang et al., 2016*), this framework was designed to simulate how an agent constructed the world model on gravity through agent-environment interactions. Specifically, a world model with undifferentiated directions of gravity generated a prediction on the stability of an object, and the mismatches between the prediction and the observation of the object from the physical world were used to fine-tune the distribution of the directions in the world model. This process is similar to how humans update their internal knowledge by comparing simulated expectations (*Hegarty, 2004*; *Ullman et al., 2017*) with actual observations (*Baillargeon, 2004*; *Baillargeon, 1994*; *Kotovsky and Baillargeon, 2000*). After several generations of error minimization, a Gaussian distribution of gravity's direction with the vertically downward direction as the maximum likelihood was similar to that observed in the human world model. Interestingly, when the physical worlds that the agent interacted with changed their appearance with stacks of different heights, the world models maintained their general patterns, but the stochastic representation of gravity's direction changed accordingly. This finding not only supports the robustness of the active inference (*Hegarty, 2004*; *Ullman et al., 2017*), which efficiently encodes critical features under different physical worlds, but also resonates with the idea that intelligence develops under the constraints of the physical world. Taken together, the finding from the RL framework implies that the world model on

gravity in humans may also be constructed in the same way, possibly through the mechanism of the predictive coding in a generative process (*Friston, 2018*; *Huang and Rao, 2011*).

Our world model on gravity provides an example of the world model theory that emphasizes the predictive nature of generative neural networks implemented with a priori knowledge of the physical world (*Friston et al., 2021*; *Land, 2014*; *Matsuo et al., 2022*). In contrast to traditional discriminative neural networks that learn statistical patterns for stability from gigantic amounts of labeled stacks, generative models equipped with the physics laws governing the physical world rely much less on experience. Importantly, the stochastic feature of the model further enhances the efficiency by balancing accuracy and speed, which improves our chances of better survival (*Cosmides and Tooby, 1997*) and adaptation to novel environments (e.g., astronauts in outer space; *Wang et al., 2022*). Indeed, the close link between human cognition and the physical world through interaction may shed light on the development of a new generation of AI with human-like intelligence that can work flexibly in open-ended environments (*Marcus, 2020*; *Marcus, 2018*).

## Methods

### Creating stacks with different configurations

We designed a block-stacking procedure in a physical simulation platform (PyBullet) to generate stacks with different configurations. All stacks used in this study were generated using this procedure with the same parameters listed below.

The block-stacking procedure includes three steps (*Figure 1—figure supplement 1a*): (1) defining the designated area, (2) stacking blocks, and (3) fine-tuning block positions. The first step is to designate a restricted place area. All blocks of a stack were required to be placed within the designated area. The designated area controls the aggregation level of blocks, with a small area clustering blocks closer than a large area. The designated area is determined by two horizontal parameters $x$ and $y$, which separately represent the size of the area in two horizontal directions. Therefore, when the block number is fixed, a smaller area in general constructs a higher stack. After designating the area, in step 2 we stacked blocks in random horizontal positions within the area one by one. If no block was positioned under a new block, the new block would be directly placed on the ground; otherwise, it would stack on the positioned block. The horizontal position of each block was independently sampled from a uniform distribution, with lower and upper bounds being -$x$ and +$x$, or -$y$ and +$y$ separately ($x$ and $y$ were all independently sampled from a uniform distribution $U(0.2, 2.0)$). The first two steps allow us to generate a large number of configurations within the designated area, which is the only restriction of the block-stacking procedure. To better control the physical stability of each stack, in step 3 we fine-tuned blocks in the stack by adjusting overlaps between every neighboring one, which was randomly sampled from a uniform distribution $U(0.2, 0.8)$. Smaller overlap between neighboring blocks is more likely to construct unstable stacks, whereas more extensive overlap results in more stable stacks. The overlap of neighboring blocks without contact is set to 0. Note that the overlap between neighboring blocks is not the only factor determining a stack's stability, and step 3 is used to generate stacks without consuming too many computational resources.

The size of each block has a 3D aspect ratio of 3:1:1 (length:width:height), with an arbitrary unit of 1.2:0.4:0.4. This constitutes three types of blocks (length, width, or height is 1.2, respectively, see *Figure 1—figure supplement 1b*). Each block of a stack was randomly selected as one of the three types of blocks. The mass of each block is set to 0.2 kg, and the friction coefficients and the coefficients of restitution between blocks are set to 1 and 0, respectively.

### Estimating the stability of a stack

The stability of a stack was obtained by a rigid-body forward simulation under the natural gravity environment (i.e., NGS). The direction of the natural gravity points downward (i.e., $\vec{G} = (0, 0, -9.8)$), and all blocks of a stack are affected by the same gravity. Gravity is the only factor for changing the state of each block, and no external force is added during the simulation. Within each simulation, we recorded 500 simulation stages. In each stage, the center position of each block was collected to measure the stability of the stack. If the position of any block does not change during the simulation, the stack is considered stable, otherwise unstable. We formulate the stack's state according to the below criteria:

$$Stable: \ \forall t \wedge \forall m, \ |P_{tm} - P_{0m}| < \varepsilon$$
$$Unstable: \ \exists t \vee \exists m, \ |P_{tm} - P_{0m}| > \varepsilon \tag{1}$$

where $t$ is a simulation stage, $m$ is the block number of a stack, $P_{tm}$ is the position of the block $m$ at stage $t$, and $\varepsilon$ is the just noticeable difference (i.e., j.n.d) of the perception, which is set to 0.01.

The stability of a stack is further calculated by measuring the proportion of displaced blocks, which is formulated as follows:

$$Stability = \frac{\sum\limits_{m=1}^{M} I\left(|P_{Tm} - P_{0m}| < \varepsilon\right)}{M} \tag{2}$$

where $M$ is the total number of blocks of a stack, and $T$ is the final stage of the simulation (i.e., $T = 500$). $\mathbb{I}(\cdot) = 1$ when $|P_{Tb} - P_{0b}| < \varepsilon$, which denotes that the stack is stable.

## Measuring participants' sensitivity to gravity's direction

We decomposed gravity's direction into three independent components (*Figure 1b*):

$$G_x = g \sin\theta \cos\varphi$$
$$G_y = g \sin\theta \sin\varphi$$
$$G_z = g \cos\theta \tag{3}$$

where $g$ is the magnitude of gravity ($g = 9.8$), which was fixed in this study. $\theta$ represents the vertical component, $\varphi$ represents the horizontal component, and $x$, $y$, and $z$ are three mutually perpendicular axes. The direction of the gravity was determined by the angle pair $(\theta, \varphi)$, where $\theta$ affects the extent of the collapse, and $\varphi$ affects the orientation of the collapse. When $\theta$ is 0, gravity's direction is vertical.

We performed a psychophysics experiment to measure humans' sensitivity to gravity's direction. In this experiment, 10 participants (five females, age range: 21–28 y) from Tsinghua University were recruited to finish four runs of the behavioral experiment, which measured their ability to detect the abnormality of stacks' collapse trajectories. The experiment was approved by the Institutional Review Board of Tsinghua University (2022 no. 34), and informed consent was obtained from all participants before the experiment.

The collapse trajectory of a stack was solely determined by gravity with different directions, where larger values of $\theta$ and $\varphi$ made the trajectories more abnormal. A pilot experiment showed that almost all $\theta_s$ greater than 45° made the collapse trajectory abnormal to most participants, and therefore in the experiment, $\theta$ ranges from 0° to 45° with a step of 3°. $\varphi$ ranges from 0° to 360° with a step of 24°. Therefore, $\theta$ and $\varphi$ consists of 16 values, respectively, which were randomly combined into 96 pairs of $(\theta, \varphi)$, with each value repeating six times in each run. In a trial, an unstable stack was constructed, and then the camera rotated one circle to show the 3D configuration of the stack to participants (*Figure 1—video 1*). The configuration was randomly selected from a dataset with more than 2000 unstable stacks, which was generated with the block-stacking procedure before the experiment. Each stack in the database was constructed with 10 blocks, and the color of each block was randomly rendered. There was a 1 s delay after the rotation, during which the participants were instructed to infer the collapse trajectory based on the configuration. Then, simulated gravity with a direction determined by an angle pair $(\theta, \varphi)$ was applied to the stack, and the stack started to collapse. If the collapse trajectory met participants' expectations, they were instructed to choose 'normal,' otherwise 'abnormal.' Once the judgment was made, the subsequent trial started immediately. Each trial lasts about 10 s, taking 16 min for a run.

In addition, to test if participants' sensitivity to gravity's direction is encapsulated from visual experience and task context, we flipped gravity's direction upside down by inverting the camera's view, and the rest procedure remained the same.

To calculate participants' sensitivity to gravity's direction, we converted their behavioral judgment into confidence levels about the normal trajectory, which is the percentage that a trajectory was judged as normal, which was calculated as below:

$$Ratio_{\theta, \varphi} = \frac{n_{\theta, \varphi}}{N_{\theta, \varphi}} \tag{4}$$

where $n_{\theta,\varphi}$ is the number of trajectories that were judged as 'normal' with the angle pair $(\theta, \varphi)$, and $N_{\theta,\varphi}$ is the total number of trajectories with the same angle pair. Because the angle pairs tested were a subset of all possible angle pairs, we used the average ratio along $\varphi$ as the ratio of angle pairs untested (*Figure 1c*) to acquire each participant's tuning curve. Finally, we calculated participants' sensitivity by fitting their confidence levels at different $\theta$ to a Gaussian distribution.

$$Ratio_\theta = Ae^{-\frac{\theta^2}{2\sigma^2}} \tag{5}$$

where $Ratio_\theta$ is the confidence level of $\theta$, which was calculated by averaging the confidence level along all $\varphi_s$, $A$ is the magnitude of the Gaussian curve, and $\sigma$ is the variance of the Gaussian curve. The best-fitted $\sigma$ was used to index participants' sensitivity to gravity's direction, and a larger $\sigma$ indicates a lower sensitivity.

## Measuring participants' ability on stability inference

Another group of 11 participants (five females, age range: 21–32 y) from Tsinghua University completed a behavioral experiment for judging the stability of 60 stacks. The experiment was approved by the Institutional Review Board of Tsinghua University, and informed consent was obtained from all participants before the experiment. One male participant (age: 25 y) was excluded from further analyses because his judgment showed an extremely weak correlation with the actual stability of stacks ($r_s < 0.30$ for all experimental runs) compared to the rest of the participants.

The stacks contained 26 unstable and 34 stable stacks, which were randomly interleaved in each run. The participants were instructed to judge stacks' stability on an 8-point Likert scale, with 0 referring to 'definitely unstable' and 7 to 'definitely stable.' There was no feedback after each judgment. The participants completed six runs, within which the same group of stacks was presented but the sequence, blocks' colors, and camera's perspective were all randomized. After the experiment, only two participants reported that they suspected a few stacks were repeated in different runs, but they could not locate the stacks they suspected. Besides, their behavioral performance was not significantly different from other participants.

Participants' stability judgment was rescaled to 0 and 1 to match the scale of the stacks' stability. The participants' inference bias (IB) was indexed as the difference in stability judgment between the participants and the NGS, shown as

$$IB = Stability_{human} - Stability_{NGS} \tag{6}$$

Negative IB indicates that participants tended to consider a stable stack as an unstable one.

## Estimating the stability of stacks based on the stochastic world model on gravity

The actual stability of a stack can be calculated with a one-time simulation of NGS ($\vec{G} = (0, 0, -9.8)$). In contrast, the stochastic nature of mental gravity requires a multiple-time simulation with different gravity's directions. Specifically, we first randomly sampled several angle pairs $(\theta_s, \varphi_s)$ from the Gaussian distribution of gravity's directions in humans. The distribution was the average of two distributions acquired from the real world (i.e., gravity's direction is downward) and the inverted world (the direction is upward), with angles having larger confidence levels more likely being sampled. We then applied the simulated gravity with these sampled directions to the stack and used the averaged stability with these directions as the stability of the stack estimated by the MGS. Similar to the IB between the participants and the NGS, the IB between the MGS and NGS was calculated as

$$IB = Stability_{MGS} - Stability_{NGS} \tag{7}$$

Stacks of different heights were created to investigate whether the stochastic world model on gravity results in the illusion that tall objects are considered less stable than short ones. The height of a stack was correlated with the size of the designated area, with a smaller area size corresponding to taller stacks. Therefore, we designated several square areas with different sizes. The side length of the squares ranged from 0.2 to 2.0, with an increase of 0.1. For each square, we used the block-stacking

procedure to generate 100 stable and 100 unstable stacks consisting of 10 blocks. The height of each stack was the height of the highest block.

## Investigating the origin of the stochastic world model on gravity

An RL framework was used to simulate the development of the stochastic nature of the world model on gravity. To do this, we first created stacks whose block number ranged from 2 to 15 with the block-stacking procedure, and initialized a spherical force space, where $\theta$ ranged from 0° to 180° and $\varphi$ from 0° to 360°, separately divided them into 61 sampling angles across the spherical force space (i.e., the angle density). The spherical space covered all possible force directions, with the initial probability of being sampled by the MGS identical. During the training, three angle pairs $(\theta_s, \varphi_s)$ were sampled according to the probability of the spherical space, and then applied to a stack for simulating its collapse trajectory, which was divided into 500 stages. We optimized the sampling probability of gravity's direction by comparing the estimated stability (i.e., expectation) with the actual stability (i.e., observation) as a $Q$ value, with a higher $Q$ value suggesting that the sampled gravity's direction more likely mismatched the actual gravity's direction. The $Q$ value was calculated as

$$Q = \frac{\sum_{m=1}^{M} \mathbb{I}\left(\left|P_{m,(\theta,\varphi)} - P_m\right| < \varepsilon\right)}{M} \tag{8}$$

where $P_{m,(\theta,\varphi)}$ is the final position of block $m$ with gravity's direction $(\theta, \varphi)$, $P_m$ is the final position of block $m$ with NGS, $M$ is the block number of the stack, and the j.n.d. $\varepsilon$ is set to 0.01. The mismatch between the expectation and the observation was used to update the sampling probability of the angle pair using a temporal difference optimization

$$W_{\theta,\varphi} \leftarrow W_{\theta,\varphi} + \gamma \left(Q - W_{\theta,\varphi}\right) \tag{9}$$

where $\gamma = 0.15$ is the learning rate. This process was iterated to update the sample probability of angle pairs $(\theta_s, \varphi_s)$ until the training stopped. The angle density and learning rate are two factors that affect the learning speed. A larger angle density prolongs the time to reach convergence but enables a more detailed force space; a higher learning rate accelerates convergence but incurs larger variance during training. To balance speed and convergence, we utilized 100,000 configurations for the training.

## Evaluating the ecological advantage of the model

To investigate how the world model on gravity balances response accuracy and speed, we trained a linear classifier (i.e., logistic regression) to model humans' decision-making process at different simulation stages. During the simulation, the same stack was separately simulated using the NGS and MGS, and we collected the position coordinates of all blocks at each stage. Differences in the positions of the blocks between the intermediate stage and the initial stage provided information about the stability of a stack, with more displaced blocks suggesting the lower stack's stability. As the simulation proceeded, differences in position gradually accumulated for unstable stacks, otherwise unchanged for stable stacks. The linear classifier was trained to judge whether a stack is stable with differences in position as inputs.

We used the block-stacking procedure to create stacks consisting of 2–10 blocks and estimated their stabilities with the NGS for simulation in 500 stages. For each block number, there were 100 stable and 100 unstable stacks to train the linear classifier, and its prediction accuracy was measured with another group of 100 stable and 100 unstable stacks at every simulation stage.

The difference in positions of each block between the intermediate and initial stages was used as the input of the linear classifier. Specifically, we collected all vertex positions of a block during the simulation to acquire the difference in position, which included eight coordinate points for each block in each stage. We did not collect the central position as previously used in the stability estimation simply because it did not provide information on the shape and size of the block. We separately performed the simulation using the MGS and NGS, calculated the difference in position between the intermediate stage and the initial stage, and then flattened the difference to generate 24 position features for each block (i.e., eight positions per block in three-dimensional space). Therefore, for a 10-block stack as an example, 240 position features were prepared as the input of the linear classifier.

Prediction accuracy at each stage was estimated by evaluating whether a stack tested was stable with the MGS or with the NGS. The highest accuracy in the whole simulation stages was used as the prediction accuracy. Accordingly, the first simulation stage to reach the maximum accuracy provided information on response speed: reaching the maximum accuracy with a smaller number of stages indicates the classifier model accomplishes stability inference in a shorter amount of time (i.e., quick response). Therefore, we measured the response speed by estimating the steps to reach the accuracy plateau

$$Time = \frac{\hat{t}}{T}$$
$$\hat{t} = \arg\max_{t} Accuracy_t$$

(10)

where $Accuracy_t$ is the accuracy of stage $t$, $\hat{t}$ is the stage that a linear classifier acquires the maximum accuracy for the first time, and $T$ is the total stage number of each simulation ($T = 500$). Higher values indicate longer response time (i.e., slower response). Finally, the efficiency of the stability inference, which is the balance between accuracy and speed, was found by dividing the prediction accuracy by the response time.

$$Efficiency = \frac{Accuracy}{Time}$$

(11)

## Acknowledgements

We thank all the members of the Liu Lab for their valuable comments.

## Additional information

### Funding

| Funder | Grant reference number | Author |
|---|---|---|
| Beijing Municipal Science & Technology Commission and Administrative Commission of Zhongguancun Science Park | Z221100002722012 | Jia Liu |
| Tsinghua University Guoqiang Institute | 2020GQG1016 | Jia Liu |
| Tsinghua University Qiyuan Laboratory | | Jia Liu |
| Beijing Academy of Artificial Intelligence | | Jia Liu |
| The Shimu Tsinghua Scholar Program | | Taicheng Huang |

The funders had no role in study design, data collection and interpretation, or the decision to submit the work for publication.

### Author contributions

Taicheng Huang, Conceptualization, Resources, Data curation, Software, Formal analysis, Validation, Investigation, Visualization, Methodology, Writing - original draft, Project administration, Writing - review and editing; Jia Liu, Conceptualization, Supervision, Funding acquisition, Writing - original draft, Project administration, Writing - review and editing

### Author ORCIDs

Taicheng Huang (iD) http://orcid.org/0000-0002-7670-1124
Jia Liu (iD) http://orcid.org/0000-0003-0383-0934

## Ethics

Human subjects: The experiment was approved by the Institutional Review Board of Tsinghua University (2022 No. 34), and informed consent was obtained from all participants before the experiment.

Reviewer #2 (Public Review): https://doi.org/10.7554/eLife.88953.3.sa1

Author response https://doi.org/10.7554/eLife.88953.3.sa2

# Additional files

## Supplementary files

• MDAR checklist

## Data availability

All analyses are included in the manuscript. The data and code are freely available from Figshare (https://doi.org/10.6084/m9.figshare.25591104.v1).

The following dataset was generated:

| Author(s) | Year | Dataset title | Dataset URL | Database and Identifier |
|---|---|---|---|---|
| Huang T, Liu J | 2024 | A stochastic world model on gravity for stability inference | https://doi.org/10.6084/m9.figshare.25591104.v1 | figshare, 10.6084/m9.figshare.25591104.v1 |

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

## Appendix 1

### Estimate the lower bound of the possible number of configurations

A configuration is a structure composed of several contact blocks. To simplify the computation of estimating the number of possible configurations, here we constrained the shape of blocks and the position where the blocks were placed.

The shape constraint: the blocks used to form a configuration are all uniform rectangular blocks with the same aspect ratio.

The position constraint: only one block is allowed to be placed on the same layer of the configuration.

Thus, the problem is then simplified to estimate the possible number of configurations when only one rectangular block with the aspect ratio of (i.e., **the shape constraint**) is allowed to place in one layer (i.e., **the position constraint**). Note that the constraints significantly reduce the number of estimated configurations.

We illustrated our solution by starting with a simple case: the aspect ratio of blocks is $\alpha : \alpha : \alpha$.

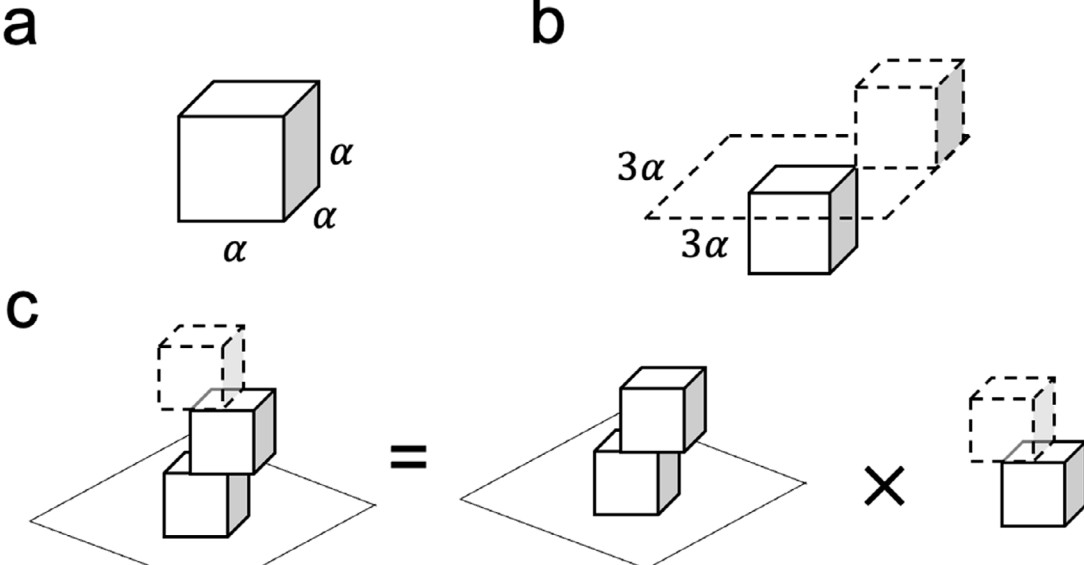

**Appendix 1—figure 1.** An illustration of the procedure to estimate the possible number of configurations when blocks have an aspect ratio of $\alpha : \alpha : \alpha$. (**a**) The cubic block with the length, width and height are $\alpha$. (**b**) Constructing a configuration by stacking two cubic blocks. The upper block could only be placed within a $3\alpha \times 3\alpha$ area to guarantee contact with the lower block. (**c**) A three-block configuration can be viewed as stacking a cubic block on a two-block configuration.

### The condition when the aspect ratio of blocks is $\alpha : \alpha : \alpha$

The block with the aspect ratio of is a cube (*Appendix 1—figure 1a*). The side length of the cube is defined as α. Consider a configuration with two stacking blocks, the upper block needs to be placed in a $3\alpha \times 3\alpha$ area to ensure contact with the bottom block (*Appendix 1—figure 1b*). To estimate the possible number of this simple situation, we defined a visual acuity $v$, which is the minimum resolution to distinguish two stacks (i.e., j.n.d.). Note that $v$ is a small value and here we set it as $v$ = 0.01 to match the minimal position difference for stability estimation in the simulation platform (please see Methods). Therefore, the possible number of the configuration containing two cubic blocks is

$$N_{C2} = (\frac{2\alpha}{v})^2 \tag{A1}$$

Where $N_{C2}$ indicates the possible number of configurations containing two cubic blocks.

We further consider the situation with more cubic blocks. For a stack that contains three cubic blocks, it can be viewed as placing a cubic block on a two-block stack (*Appendix 1—figure 1c*).

Therefore, the total possible number of configurations is the multiplication of two two-block configurations, which is formulated as

$$N_{C3} = N_{C2} \times N_{C2} = N_{C2}{}^2$$

Similarly, the possible number of configurations for stacks containing four cubic blocks is

$$N_{C4} = N_{C3} \times N_{C2} = N_{C2}{}^3$$

Accordingly, the possible number of configurations with M cubic blocks is

$$N_{CM} = N_{c(M-1)} \times N_{C2} = \cdots = N_{C2}{}^{M-1} = \left(\frac{2\alpha}{v}\right)^{2M-2}, M \geq 2 \tag{A2}$$

Now, we have introduced the basic idea of calculating the number of configurations using a block with an $\alpha : \alpha : \alpha$ aspect ratio as a special case. Then we generalized the idea to estimate the possible number when the block is rectangular with the aspect ratio as $\alpha : \beta : \beta$.

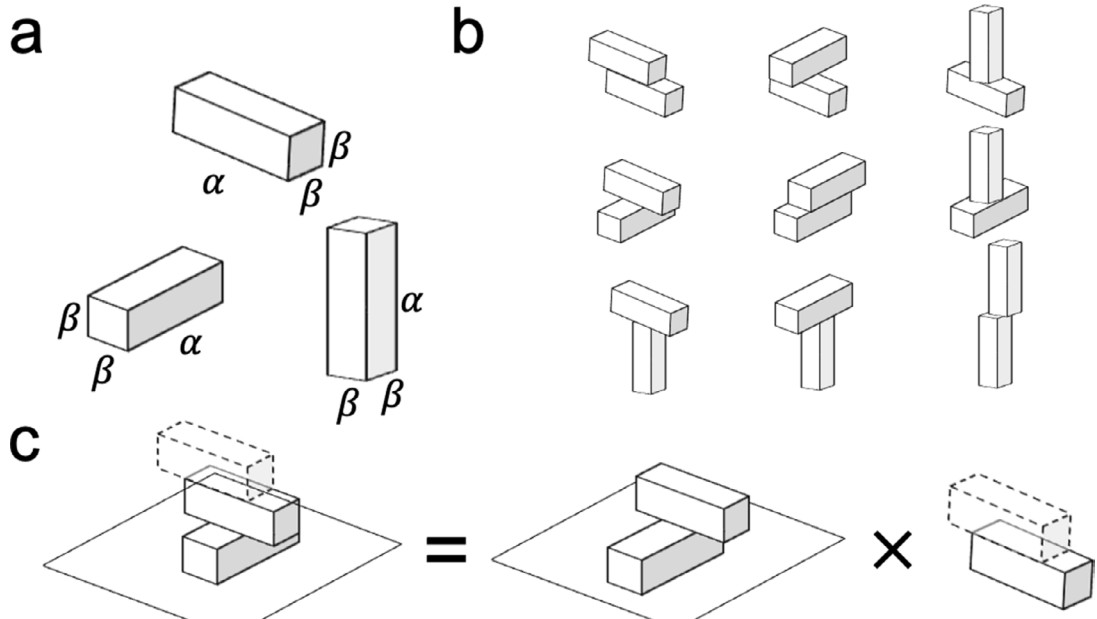

**Appendix 1—figure 2.** An illustration of the procedure to estimate the possible number of configurations when blocks have the aspect ratio of $\alpha : \beta : \beta$. (**a**) Three types of rectangular blocks with an aspect ratio of $\alpha : \beta : \beta$. (**b**) There are nine possible two-block configurations when combining blocks with an aspect ratio of $\alpha : \beta : \beta$. (**c**) A three-block configuration could be viewed as stacking a cubic block on a two-block configuration.

### The condition when the aspect ratio of blocks is $\alpha : \beta : \beta$

A block with the aspect ratio of $\alpha : \beta : \beta$ has three types, corresponding to the sides of length, width and height are and the rest sides are $\beta$ ($\alpha : \beta : \beta$, $\beta : \alpha : \beta$, and $\beta : \beta : \alpha$; see *Appendix 1—figure 2a*). For simplicity, we label the three basic blocks as A, B and C. The three types of blocks can generate 9 (i.e., $3^2$) two-block configurations in total (*Appendix 1—figure 2b*). We calculate each of the possible numbers of two-block configurations below.

$$N_{R2} = \begin{bmatrix} N_{AA} & N_{AB} & N_{AC} \\ N_{BA} & N_{BB} & N_{BC} \\ N_{CA} & N_{CB} & N_{CC} \end{bmatrix}$$

$$= \frac{1}{v^2} \begin{bmatrix} 4\alpha\beta & (\alpha+\beta)^2 & 2\beta(\alpha+\beta) \\ (\alpha+\beta)^2 & 4\alpha\beta & 2\beta(\alpha+\beta) \\ 2\beta(\alpha+\beta) & 2\beta(\alpha+\beta) & 4\beta^2 \end{bmatrix} \quad \text{(A3)}$$

The possible number of configurations for stacks containing two rectangular blocks with the aspect ratio of $\alpha : \beta : \beta$ is

$$N_{R2} = \sum N_{R2} \quad \text{(A4)}$$

For a configuration containing three blocks, it can be viewed as a block stacked on a two-block stack (*Appendix 1—figure 2c*). Therefore,

$$N_{R3} = N_{..A} + N_{..B} + N_{..C} \quad \text{(A5)}$$

Where $N_{..A}$ indicates the possible number when block A stacked at the upper layer, and each term can be expanded as below.

$$\begin{aligned} N_{..C} &= N_{.A} \times N_{AC} + N_{.B} \times N_{BC} + N_{.C} \times N_{CC} \\ N_{..A} &= N_{.A} \times N_{AA} + N_{.B} \times N_{BA} + N_{.C} \times N_{CA} \\ N_{..B} &= N_{.A} \times N_{AB} + N_{.B} \times N_{BB} + N_{.C} \times N_{CB} \end{aligned} \quad \text{(A6)}$$

Combining *Equations A4–A6*, we have

$$N_{R3} = \sum \left( \begin{bmatrix} N_{.A} & N_{.B} & N_{.C} \end{bmatrix} \times \begin{bmatrix} N_{AA} & N_{AB} & N_{AC} \\ N_{BA} & N_{BB} & N_{BC} \\ N_{CA} & N_{CB} & N_{CC} \end{bmatrix} \right)$$

And

$$\begin{bmatrix} N_{.A} & N_{.B} & N_{.C} \end{bmatrix} = \begin{bmatrix} 1 & 1 & 1 \end{bmatrix} \times \begin{bmatrix} N_{AA} & N_{AB} & N_{AC} \\ N_{BA} & N_{BB} & N_{BC} \\ N_{CA} & N_{CB} & N_{CC} \end{bmatrix}$$

Therefore,

$$N_{R3} = \sum \left( N_{R2}^2 \right) \quad \text{(A7)}$$

Following a similar logic, the possible number of configurations containing M blocks with an aspect ratio of $\alpha : \beta : \beta$ is

$$N_{RM} = \sum \left( N_{R2}^{M-1} \right), \, M \geq 2 \quad \text{(A8)}$$

## The aspect ratio of blocks is $\alpha : \beta : \gamma$

We further generalize the problem by considering the aspect ratio of blocks as $\alpha : \beta : \gamma$. This forms six different types: $\alpha : \beta : \gamma$, $\alpha : \gamma : \beta$, $\beta : \alpha : \gamma$, $\beta : \gamma : \alpha$, $\gamma : \alpha : \beta$, $\gamma : \beta : \alpha$, for each type the three proportional values corresponding to length, width and height, respectively. We label the six types of blocks as A, B, C, D, E, F, and G for simplicity.

Following the similar logic as above, different types of blocks generated 36 (i.e., $6^2$) two-block configurations in total, and the possible number of each two-block configuration is

$$N_{R2} = \begin{bmatrix} N_{AA} & N_{AB} & N_{AC} & N_{AD} & N_{AE} & N_{AF} \\ N_{BA} & N_{BB} & N_{BC} & N_{BD} & N_{BE} & N_{BF} \\ N_{CA} & N_{CB} & N_{CC} & N_{CD} & N_{CE} & N_{CF} \\ N_{DA} & N_{DB} & N_{DC} & N_{DD} & N_{DE} & N_{DF} \\ N_{EA} & N_{EB} & N_{EC} & N_{ED} & N_{EE} & N_{EF} \\ N_{FA} & N_{FB} & N_{FC} & N_{FD} & N_{FE} & N_{FF} \end{bmatrix} \tag{A9}$$

$$= \frac{1}{v^2} \begin{bmatrix} 4\alpha\beta & 2\alpha(\beta+\gamma) & (\alpha+\beta)^2 & (\alpha+\beta)(\beta+\gamma) & (\alpha+\gamma)(\alpha+\beta) & 2\beta(\alpha+\gamma) \\ 2\alpha(\beta+\gamma) & 4\alpha\gamma & (\alpha+\beta)(\alpha+\gamma) & 2\gamma(\alpha+\beta) & (\alpha+\gamma)^2 & (\alpha+\gamma)(\beta+\gamma) \\ (\alpha+\beta)^2 & (\alpha+\beta)(\alpha+\gamma) & 4\alpha\beta & 2\beta(\alpha+\gamma) & 2\alpha(\beta+\gamma) & (\alpha+\beta)(\beta+\gamma) \\ (\alpha+\beta)(\beta+\gamma) & 2\gamma(\alpha+\beta) & 2\beta(\alpha+\gamma) & 4\beta\gamma & (\beta+\gamma)(\alpha+\gamma) & (\beta+\gamma)^2 \\ (\alpha+\beta)(\alpha+\gamma) & (\alpha+\gamma)^2 & 2\alpha(\beta+\gamma) & (\alpha+\gamma)(\beta+\gamma) & 4\alpha\gamma & 2\gamma(\alpha+\beta) \\ 2\beta(\alpha+\gamma) & (\alpha+\gamma)(\beta+\gamma) & (\alpha+\beta)(\beta+\gamma) & (\beta+\gamma)^2 & 2\gamma(\alpha+\beta) & 4\beta\gamma \end{bmatrix}$$

The possible number of configurations for stacks with M blocks with an aspect ratio $\alpha : \beta : \gamma$ is

$$N_{RM} = \sum \left( N_{R2}^{M-1} \right), \, M \geq 2 \tag{A10}$$

Therefore, we can estimate the possible number of configurations when only one rectangular block with the aspect ratio of $\alpha : \beta : \gamma$ is allowed to place in each layer using the formula *Equations A9; A10*.

Finally, in this study we chose blocks with an aspect ratio of 3:1:1 as building blocks for stacks whose stability was evaluated. Specifically, for stacks consisting of 10 blocks and j.n.d. of $v = 0.01$, the number of configurations can be estimated with formula *Equation A10*, which is $1.14 \times 10^{50}$.

