## [Editor Report · eLife assessment]

In this **valuable** study, the authors present findings that suggest that people do not faithfully replicate the physics of the real world but rather have a stochastic world model, specifically a stochastic representation of gravity. This contrasts with prior accounts that suggested a potentially noisy Newtonian model where the noise arises from perceptual uncertainty or (inferred) external perturbations. The experimental evidence is generally **solid**, with all experiments and model simulations being consistent with the proposed account. In the revision, the authors also added a number of control experiments that address some of the most pressing concerns of the original submission.

---

## [Referee Report · Reviewer #2 (Public Review)]

Summary:

Through a set of experiments and model simulations, the authors tested whether the commonly assumed world model of gravity was a faithful replica of the physical world. They found that participants did not model gravity as single, fixed vector for gravity but instead as a distribution of possible vectors. Surprisingly, the width of this distribution was quite large (~20 degrees). While previous accounts had suggested that this uncertainty was due to perceptual noise or an inferred external perturbation, the authors suggest that this uncertainty simply arises from a noisy distribution of the representation of gravity's direction. A reinforcement learning model with an initial uniform distribution for gravity's direction ultimately converged to a precision on the same order as the human participants, which lends support to the authors' conclusion and suggests that this distribution is learned through experience. What's more, further simulations suggest that representing gravity with such a wide distribution may balance speed and accuracy, providing a potentially normative explanation for the world model with gravity as a distribution.

Strengths:

The authors present surprising findings in a relatively straight-forward in a now classic experimental task. They provide a normative explanation based on a resource-rational framework for why people may have a stochastic world model instead of a deterministic world model. While the stochastic world model could be the result of people mentally simulating an external perturbation, the authors include several control experiments to test this possibility.

Weaknesses:

The possibility of inferred external perturbations, as opposed to a stochastic world model, is difficult to rule out. This could stem from how people interpret task instructions and it will likely take many, clever studies, to fully reconcile these two alternative accounts.

---

## [Author Response]

The following is the authors’ response to the original reviews.

We are grateful to the reviewers for their appreciation of our study and thoughtful comments. In response to the main concern raised by all reviewers regarding the potential influences of external noise factors on intuitive inference, such as external disturbances or imperfect observations, we have conducted three new experiments suggested by the reviewers. These experiments were designed to: (1) assess the influence of external forces on humans’ judgments by implementing a wall to block wind disturbances from one direction, (2) examine human accuracy in predicting the landing position of a falling ball when its trajectory is obscured, and (3) evaluate the effect of object geometry on human judgment of stability. The findings from these experiments consistently support our proposal of the stochastic world model on gravity embedded in human mind. Besides, we have also addressed the rest comments from the reviewers in a one-by-one fashion.

**Reviewer #1 (Recommendations For The Authors):**
As mentioned in the public review, I did not find it entirely convincing that the study shows evidence for a Gaussian understanding of gravity. There are two studies that would bolster this claim:

1. Replicate experiment 1, but also ask people to infer whether there was a hidden force. If people are truly representing gravity as proposed in the paper, you should get no force inferences. However, if the reason the Gaussian gravity model works is that people infer unseen forces, this should come out clearly in this study.

**Author response image 1. sa2fig1:** Wall experiment to test the impact of external forces on the measurement of stochastic gravity. (a) Experimental setting. We replicated the original setup with the addition of a wall implemented on one side. Left: the overall experimental scene; Right, the scene shown to participants. (b) Human behaviors. Three participants conducted this experiment, and their responses consistently showed normal distributions without any skewness, suggesting that their judgments were not affected by the presence of the wall. These results support our claim that humans’ judgments on stability were not affected by potential concerns regarding external forces.

R1: We thank the reviewer for this suggestion. To directly test whether participants’ judgments were influenced by their implicit assumptions about external forces, we duplicated the original experimental setup with the addition of a wall implemented on one side (Supplementary Figure 4A). Before the start of the experiment, we explicitly informed the participants that the wall was designed to block wind, ensuring that any potential wind forces from the direction of the wall would not influence the collapse. If participants’ judgments were affected by external noise, we would expect to observe a skewed angle distribution. Contrary to this prediction, our results showed a normal distribution across all three participants tested (1 female; ages: 24-30), similar to the experiment without the wall (Supplementary Figure 4B). Therefore, the stochastic nature of intuitive inference on objects’ stability is embedded in the mind, not shaped by external forces or explicit instructions.

This new experiment has been added to the revised manuscript

Line 166-168: “…, and remained unchanged with the addition of a wall on one side to block potential external disturbances from wind (Supplementary Figure 4).”

(2) Similarly, you can imagine a simple study where you drop an object behind a floating occluder and you check where people produce an anticipatory fixation (i.e., where do they think the object will come out?). If people have a stochastic representation of gravity, this should be reflected in their fixations. But my guess is that everyone will look straight down.

**Author response image 2. sa2fig2:** Trajectory experiment to test the stochastic nature of gravity represented in the mind. (a) Experiment design. In this experiment, participants were required to use a mouse to determine the landing point of a parabolic trajectory (marked by the green dot), obscured by a grey rectangle. Note that the parabolic trajectory was determined only by gravity, and no external disturbances were introduced. The parameters used in this experiment are detailed in the upper right corner. (b) Predictive errors from three participants. The predictive errors from all three participants conform to Gaussian distributions with non-negligible variances. These results suggest the notion of an inherent stochastic property of gravity represented in the mind.

R2: We thank the reviewer for suggesting this thought experiment. However, when predicting the landing point of a falling object, participants may rely more on learned knowledge that an unimpeded object continues to fall in a straight line, rather than drawing on their intuitive physics. To avoid this potential confounding factor, we designed a similar experiment where participants were asked to predict the landing point of a parabolic trajectory, obscured by an occluder (Author response image 2A). In each trial, participants used a mouse (clicking the left button) to predict the landing point of each parabolic trajectory, and there were 100 trials in total. This design not only limits the impact of direct visual cues but also actively engages the mental simulation of intuitive physics. All three participants (1 female; ages: 24-30) were unable to accurately predict the landing points of the trajectories, and the predictive errors conformed to Gaussian distributions with different variances (Author response image 2B). Therefore, this new experiment confirms the stochastic nature of intuitive physics.

(3) I believe the correct alternative model should be the one that has uncertainty over unseen forces, which better captures current proposals in the field, and controls for the amount of uncertainty in the models.

R3: We thank the reviewers for the above-mentioned suggestions, and the findings from these two new experiments reinforce our proposal regarding the inherent stochastic characteristic of how the mind represents gravity.

(4) I was not convinced that the RL framework was set up correctly to tackle the questions it claims to tackle. What this shows is that you can evolve a world model with Gaussian gravity in a setup that has no external perturbations. That does not imply that that is how humans evolved their intuitive physics, particularly when creatures have evolved in a world full of external perturbations. Showing that when (1) there are hidden perturbations, and (2) these perturbations are learnable, but (3) the model nonetheless just learns stochastic gravity, would be a more convincing result.

R4: We completely agree with the reviewer that the RL framework serves primarily as a theoretic model to explain the stochastic nature of the world model on gravity, rather than as a demonstration of the developmental origins of intuitive physics abilities. The genesis of such abilities is multifaceted and unlikely to be fully replicated through a simple simulation like RL. Therefore, the purpose of incorporating the RL framework in our study is to demonstrate that external perturbances are not necessary for the development of a stochastic representation of gravity. In fact, introducing additional external noise into the RL framework likely heightens the uncertainty in learning gravity’s direction, potentially amplifying, rather than diminishing, the stochastic nature of mental gravity.

In revision, we have clarified the role of the RL framework

Line 265-277: “While the cognitive impenetrability and the self-consistency observed in this study, without resorting to an external perturbation, favor the stochastic model over the deterministic one, the origin of this stochastic feature of the world model is unclear.

Here we used a reinforcement learning (RL) framework to unveil this origin, because our intelligence emerges and evolves under the constraints of the physical world. Therefore, the stochastic feature may emerge as a biological agent interacts with the environment, where the mismatches between external feedback from the environment and internal expectations from the world model are in turn used to fine-tune the world model (Friston et al., 2021; MacKay, 1956; Matsuo et al., 2022). Note that a key aspect of the framework is determining whether the stochastic nature of the world model on gravity emerges through this interaction, even in the absence of external noise.”

(5) Some comments on the writing:The word 'normality' is used to refer to people's judgments about whether a tower collapsed looked 'normal'. I was a bit confused by this because normality can also mean 'Gaussian' and the experiments are also sampling from Gaussian distributions. There were several points where it took me a second to figure out which sense of 'normality' the paper was using. I would recommend using a different term.

R5: We are sorry for the confusion. In revision, the term “normality” has been replaced with “confidence level about normal trajectory”.

(6) One small comment is that Newton's laws are not a faithful replica of the "physical laws of the world" they are a useful simplification that only works at certain timescales. I believe some people propose Newtonian physics as a model of intuitive physics in part because it is a rapid and useful approximation of complex physical systems, and not because it is an untested assumption of perfect correspondence.

R6: We are sorry for the inaccurate expression. We have revised our statements in the manuscriptLine 15-16: “We found that the world model on gravity was not a faithful replica of the physical laws, but instead encoded gravity’s vertical direction as a Gaussian distribution.”

(7) Line 49-50: Based on Fig 1d, lower bound of possible configurations for 10 blocks is ~17 in log-space, which is about 2.5e7. But the line here says it's 3.72e19, which is much larger. Sorry if I am missing something.

R7: We thank the reviewer to point out this error. We re-calculated the number of possible configurations using the formula (3) in the appendix, and the number of configurations with 10 blocks is:NR2=1v2[4αβ(α+β)22β(α+β)(α+β)24αβ2β(α+β)2β(α+β)2β(α+β)4β2]=10.012[1216816128884]=[1200001600008000016000012000080000800008000040000]

Thus,NR10=∑(NR210−1)=Σ[1200001600008000016000012000080000800008000040000]9=1.14×1050

This estimated number is much larger than that in our previous calculation, which has been corrected in the revised text.

Line 827-829: “(d) The lower bound of configurations’ possible number and the number of blocks in a stack followed an exponential relationship with a base of 10. The procedure can create at least 1.14×1050 configurations for stacks consisting of 10 blocks.”

Line 49-50: “… but the universal cardinality of possible configurations is at least 1.14×1050 (Supplementary Figure 1), …”

Line 1017-1018: “… the number of configurations can be estimated with formula (9), which is 1.14×1050.”

(8) Lines 77-78: "A widely adopted but not rigorously tested assumption is that the world model in the brain is a faithful replica of the physical laws of the world." This risks sounding like you are asserting that colleagues in the field do not rigorously test their models. I think you meant to say that they did not 'directly test', rather than 'rigorously test'. If you meant rigorous, you might want to say more to justify why you think past work was not rigorous.

R8: We apologize for the inappropriate wording, the sentence has been revised and we illustrate the motivation more comprehensively in the revised text,

Line 76-92: “A prevailing theory suggests that the world model in the brain accurately mirrors the physical laws of the world (Allen et al., 2020; Battaglia et al., 2013; Zhou et al., 2022). For example, the direction of gravity encoded in the world model, a critical factor in stability inference, is assumed to be straight downward, aligning with its manifestation in the physical world. To explain the phenomenon that tall and thin objects are subjectively perceived as more unstable compared to short and fat ones (Supplementary Figure 2), external noise, such as imperfect perception and assumed external forces, is introduced to influence the output of the model. However, when the brain actively transforms sensory data into cognitive understanding, these data can become distorted (Kriegeskorte and Douglas, 2019; Naselaris et al., 2011), hereby introducing uncertainty into the representation of gravity’s direction. In this scenario, the world model inherently incorporates uncertainty, eliminating the need for additional external noise to explain the inconsistency between subjective perceptions of stability and the actual stability of objects. Note that this distinction of these two theories is nontrivial: the former model implies a deterministic representation of the external world, while the latter suggests a stochastic approach.”

(9) Lines 79-84 States that past models encode gravity downward. It then says that *alternatively* there is consensus that the brain uses data from sensory organs and adds meaning to them. I think there might be a grammatical error here because I did not follow why saying there is 'consensus' on something is a theoretical alternative. I also had trouble following why those two statements are in opposition. Is any work on physics engines claiming the brain does not take data from sensory organs and add meaning to them?

R9: We are sorry for the confusion. Here we intend to contrast the deterministic model (i.e., the uncertainty comes from outside the model) with the stochastic model (i.e., the uncertainty is inherently built into the model). In revision, we have clarified the intention. For details, please see R8.

(10) Lines 85-88: Following on the sentence above, you then conclude that the representation of the world may therefore not be the same as reality. I did not understand why this followed. It seems you are saying that, because the brain takes data from sensory organs, therefore its representations may differ from reality.

R10: Again, we are sorry about the confusion. Please see the revised text in R8.

(11) Lines 190-191: I had trouble understanding this sentence. I believe you are missing an adjective to clarify that participants were more inclined to judge *taller* stacks as more likely to collapse.

R11: We are sorry for the confusion. What we intended to state here is that participants’ judgment was biased, showing a tendency to predict a collapse for stacks regardless of their actual stability. We have revised this confusing sentence in the revision.Line 202–204: “However, the participants showed an obvious bias towards predicting a collapse for stacks regardless of their actual stability, as the dots in Fig 2b are more concentrated on the lower side of the diagonal line.”

(12) Line 201: I don't think it's accurate to say that MGS "perfectly captured participants' judgments" unless the results are actually perfect.

R12: We agree, and in revision we have toned down the statementLine 213–214: “…, the MGS, in contrast to the NGS, more precisely reflected participants’ judgments of stability …”

**Reviewer #2 (Recommendations For The Authors):**
I think this is an impressive set of experiments and modeling work. The paper is nicely written and I appreciate the poetic license the authors took at places in the manuscript. I only have clarification points and suggest a simple experiment that could lend further support to their conclusions.

1. In my opinion, the impact of this work is twofold. First, the suggestion that gravity is represented as a distribution of the world and not a result of (inferred) external perturbations. Second, that the distribution is advantageous as it balances speed and accuracy, and lessens computational processing demands (i.e., number of simulations). The second point here is contingent on the first point, which is really only supported by the RL model and potentially the inverted scene condition. I am somewhat surprised that the RL model does not converge on a width much smaller than ~20 degrees after 100,000 simulations. From my understanding, it was provided feedback with collapses based on natural gravity (deterministically downward). Why is learning so slow and the width so large? Could it be the density of the simulated world model distribution? If the model distribution of Qs was too dense, then Q-learning would take forever. If the model distribution was too sparse, then its final estimate would hit a floor of precision. Could the authors provide more details on the distribution of the Qs for the RL model?

**Author response image 3. sa2fig3:** RL learning curves as a function of θ angle with different sampling densities and learning rates. Learning rates were adjusted to low (a), intermediate (b) and high (c) settings, while sampling densities were chosen at four levels: 5x5, 11x11, 31x31, and 61x61 shown from the left to the right. Two key observations emerged from the simulations as the reviewer predicted. First, higher learning rates resulted in a more rapid decline in learning curves but introduced larger variances. Second, increased sampling density necessitated more iterations for convergence. Note that in all simulations, we limited the iterations to 1,000 times (as opposed to 100,000 times reported in the manuscript) to demonstrate the trend without excessive computational demands.

R1: To illustrate the distribution of the Q-values for the RL model, we re-ran the RL model with various learning rates and sampling densities (Author response image 3). These results support the reviewer’s prediction that higher learning rates resulted in a more rapid decline in learning curves but introduced larger variances, and increased sampling density requires more iterations for convergence.

This simulation also elucidates the slower learning observed in the experiment described in the text, where the force sphere was divided into 61x61 angle pairs, and the learning rate was set to 0.15. This set of parameters ensured convergence within a reasonable brief timeframe while maintaining high-resolution force assessments.

Besides, the width of the Gaussian distribution is mainly determined by the complexity of stacks. As shown in Figure 3c and Supplementary Figure 9, stacks with fewer blocks (i.e., less complex) caused a larger width, whereas those with more blocks resulted in a narrower spread. In the study, we used a collection of stacks varying from 2 to 15 blocks to simulate the range of stacks humans typically encounter in daily life.

In revision, we have incorporated these insights suggested by the reviewer to clarify the performance of the RL framework:

Line 634-639: “The angle density and learning rate are two factors that affect the learning speed. A larger angle density prolongs the time to reach convergence but enables a more detailed force space; a higher learning rate accelerates convergence but incurs larger variance during training. To balance speed and convergence, we utilized 100,000 configurations for the training.”

Line 618-619: “…, separately divided them into 61 sampling angles across the spherical force space (i.e., the angle density).”

(2) Along similar lines, the authors discuss the results of the inverted science condition as reflecting cognitive impenetrability. However, do they also interpret it as support for an intrinsically noisy distribution of gravity? I would be more convinced if they created a different scene that could have the possibility of affecting the direction of an (inferred) external perturbation - a previously held explanation of the noisy world model. For example, a relatively simple experiment would be to have a wall on one side of the scene such that an external perturbation would be unlikely to be inferred from that direction. In the external perturbation account, phi would then be affected resulting in a skewed distribution of angle pairs. However, in the authors' stochastic world model phi would remain unaffected resulting in the same uniform distribution of phi the authors observed. In my opinion, this would provide more compelling evidence for the stochastic world model.

**Author response image 4. sa2fig4:** Wall experiment to test the impact of external forces on the measurement of stochastic gravity. (a) Experimental setting. We replicated the original setup with the addition of a wall implemented on one side. Left: the overall experimental scene; Right, the scene shown to participants. (b) Human behaviors. Three participants conducted this experiment, and their responses consistently showed normal distributions without any skewness, suggesting that their judgments were not affected by the presence of the wall. These results support our claim that humans’ judgments on stability were not affected by potential concerns regarding external forces.

R2: We thank the reviewer for this suggestion. Following the reviewer’s concern, we designed the experiment with the addition of a wall implemented on one side (Supplementary figure 4A). We explicitly informed the participants that the wall was designed to block wind before the start of the experiment, ensuring no potential wind forces from the direction of the wall to influence the collapse trajectory of configurations. Participants need to judge if the trajectory was normal. If participants’ judgments were influenced by external noises, we would expect to observe a skewed angle distribution. However, our results still showed a normal distribution across all participants tested, consistent with the experiment without the wall (Supplementary figure 4B). This experiment suggested the stochastic nature of intuitive inference on objects’ stability is embedded in the mind, rather than shaped by external forces or explicit instructions.

We revised the original manuscript, and added this new experiment

Line 166-168: “…, and remained unchanged with the addition of a wall on one side to block potential external disturbances from wind (Supplementary Figure 4).”

(3) I didn't completely follow the authors' explanation for the taller objects illusion. On lines 229-232, the authors state that deviations from gravity's veridical direction are likely to accumulate with the height of the objects. Is this because, in the stochastic world model account, each block gets its own gravity vector that is sampled from the distribution? The authors should clarify this more explicitly. If this is indeed the author's claim, then it would seem that it could be manipulated by varying the dimensions of the blocks (or whatever constitutes an object).

R3: We are sorry for the confusion caused by the use of the term ‘accumulate’. In the study, there is only one gravity vector sampled from the distribution for the entire structure, rather than each block having a unique gravity vector. The height illusion is attributed to the fact that the center of gravity in taller objects is more susceptible to influence when gravity deviates slightly from a strictly downward direction. This is especially true for objects consisting of multiple blocks stacked atop one another. In revision, we have removed the confusing term ‘accumulate’ for clarification.

Line 242-244: “…, because the center of gravity in taller objects is more susceptible to influence when gravity deviates slightly from a strictly downward direction during humans’ internal simulations.”

(4) The authors refer to the RL simulations as agent-environment interactions, but in reality, the RL model does not interact with the blocks. Would experience-dependent or observation be more apropos?

R4: We completely agree. Indeed, the RL model did not manipulate stacks; rather, it updated its knowledge of natural gravity based on the discrepancies between the RL model’s predictions and observed outcomes. In revision, we have removed the confusing term ‘agent-environment interactions’ and clarified its intended meaning.

Line 19-22: “Furthermore, a computational model with reinforcement learning revealed that the stochastic characteristic likely originated from experience-dependent comparisons between predictions formed by internal simulations and the realities observed in the external world, …”

**Reviewer #3 (Public Review):**
(1) In spite of the fact that the Mental Gravity Simulation (MGS) seems to predict the data of the two experiments, it is an untenable hypothesis. I give the main reason for this conclusion by illustrating a simple thought experiment. Suppose you ask subjects to determine whether a single block (like those used in the simulations) is about to fall. We can think of blocks of varying heights. No matter how tall a block is, if it is standing on a horizontal surface it will not fall until some external perturbation disturbs its equilibrium. I am confident that most human observers would predict this outcome as well. However, the MSG simulation would not produce this outcome. Instead, it would predict a non-zero probability of the block to tip over. A gravitational field that is not perpendicular to the base has the equivalent effect of a horizontal force applied on the block at the height corresponding to the vertical position of the center of gravity. Depending on the friction determined by the contact between the base of the block and the surface where it stands there is a critical height where any horizontal force being applied would cause the block to fall while pivoting about one of the edges at the base (the one opposite to where the force has been applied). This critical height depends on both the size of the base and the friction coefficient. For short objects this critical height is larger than the height of the object, so that object would not fall. But for taller blocks, this is not the case. Indeed, the taller the block the smaller the deviation from a vertical gravitational field is needed for a fall to be expected. The discrepancy between this prediction and the most likely outcome of the simple experiment I have just outlined makes the MSG model implausible. Note also that a gravitational field that is not perpendicular to the ground surface is equivalent to the force field experienced by the block while standing on an inclined plane. For small friction values, the block is expected to slide down the incline, therefore another prediction of this MSG model is that when we observe an object on a surface exerting negligible friction (think of a puck on ice) we should expect that object to spontaneously move. But of course, we don't, as we do not expect tall objects that are standing to suddenly fall if left unperturbed. In summary, a stochastic world model cannot explain these simple observations.

**Author response image 5. sa2fig5:** Differentiating Subjectivity from Objectivity. In both Experiment 1 (a) and Experiment 2 (b), participants were instructed to determine which shape appeared most stable. Objectively, in the absence of external forces, all shapes possess equal stability. Yet, participants typically perceived the shape on the left as the most stable because of its larger base area. The discrepancy between objective realities and subjective feelings, as we propose, is attributed to the human mind representing gravity’s direction as a Gaussian distribution, rather than as a singular value pointing directly downward.

R1: We agree with the reviewer that objects will remain stable until disturbed by external forces. However, in many cases, this is a clear discrepancy between objective realities and subjective feelings. For example, electromagnetic waves associated with purple and red colors are the farthest in the electromagnetic space, yet purple and red are the closest colors in the color space. Similarly, as shown in Supplementary Figure 4, in reality all shapes possess equal stability in the absence of external forces. Yet, humans typically perceive the shape on the left as more stable because of its larger base area. In this study, we tried to explore the mechanism underlying this discrepancy by proposing that the human mind represents gravity’s direction as a Gaussian distribution, rather than as a singular value pointing directly downward.

In revision, we have clarified the rationale of this study

Line 76-98: “A prevailing theory suggests that the world model in the brain accurately mirrors the physical laws of the world (Allen et al., 2020; Battaglia et al., 2013; Zhou et al., 2022). For example, the direction of gravity encoded in the world model, a critical factor in stability inference, is assumed to be straight downward, aligning with its manifestation in the physical world. To explain the phenomenon that tall and thin objects are subjectively perceived as more unstable compared to short and fat ones (Supplementary Figure 2), external noise, such as imperfect perception and assumed external forces, is introduced to influence the output of the model. However, when the brain actively transforms sensory data into cognitive understanding, these data can become distorted (Kriegeskorte and Douglas, 2019; Naselaris et al., 2011), hereby introducing uncertainty into the representation of gravity’s direction. In this scenario, the world model inherently incorporates uncertainty, eliminating the need for additional external noise to explain the inconsistency between subjective perceptions of stability and the actual stability of objects. Note that this distinction of these two theories is nontrivial: the former model implies a deterministic representation of the external world, while the latter suggests a stochastic approach. Here, we investigated these two alternative hypotheses regarding the construction of the world model in the brain by examining how gravity’s direction is represented in the world model when participants judged object stability.”

(2) The question remains as to how we can interpret the empirical data from the two experiments and their agreement with the predictions of the stochastic world model if we assume that the brain has internalized a vertical gravitational field. First, we need to look more closely at the questions posed to the subjects in the two experiments. In the first experiment, subjects are asked about how "normal" a fall of a block construction looks. Subjects seem to accept 50% of the time a fall is normal when the gravitational field is about 20 deg away from the vertical direction. The authors conclude that according to the brain, such an unusual gravitational field is possible. However, there are alternative explanations for these findings that do not require a perceptual error in the estimation of the direction of gravity. There are several aspects of the scene that may be misjudged by the observer. First, the 3D interpretation of the scene and the 3D motion of the objects can be inaccurate. Indeed, the simulation of a normal fall uploaded by the authors seems to show objects falling in a much weaker gravitational field than the one on Earth since the blocks seem to fall in "slow motion". This is probably because the perceived height of the structure is much smaller than the simulated height. In general, there are even more severe biases affecting the perception of 3D structures that depend on many factors, for instance, the viewpoint.

R2: We thank the reviewer for highlighting several potential confounding factors in our study. We address each of these concerns point-by-point:

(a) Misinterpretation of the 3D scene and motion. In Response Figure 4 shown above, there is no 3D structure, yet participants’ judgment on stability still deviated from objective realities. In addition, the introduction of 3D motion was to aid in understanding the stacks’ 3D structure. Previous studies without 3D motion have reported similar findings (Allen et al., 2020). Therefore, regardless of whether objects are presented in 2D or 3D, or in static or in motion formats, humans’ judgment on object stability appears consistent.

(b) Errors in perceived height. While there might be discrepancies between perceived and simulated heights, such errors are systematic across all conditions. Therefore, they may affect the width of the Gaussian distribution but do not fundamentally alter its existence.

(c) The viewpoint. In one experiment, we inverted gravity’s direction to point upward, diverging from common daily experience. Despite this change in viewpoint, the Gaussian distribution was still observed. That is, the viewpoint appears not a key factor in influencing how gravity’s direction is represented as a Gaussian distribution in our mental world.

In summary, both our and previous studies (Allen et al., 2020; Battaglia et al., 2013) agree that humans’ subjective assessments of objects’ stability deviate from actual stability due to noise in mental simulation. Apart from previous studies, we suggest that this noise is intrinsic, rather than stemming from external forces or imperfect observations.

(3) Second, the distribution of weight among the objects and the friction coefficients acting between the surfaces are also unknown parameters. In other words, there are several parameters that depend on the viewing conditions and material composition of the blocks that are unknown and need to be estimated. The authors assume that these parameters are derived accurately and only that assumption allows them to attribute the observed biases to an error in the estimate of the gravitational field. Of course, if the direction of gravity is the only parameter allowed to vary freely then it is no surprise that it explains the results. Instead, a simulation with a titled angle of gravity may give rise to a display that is interpreted as rendering a vertical gravitational field while other parameters are misperceived. Moreover, there is an additional factor that is intentionally dismissed by the authors that is a possible cause of the fall of a stack of cubes: an external force. Stacks that are initially standing should not fall all of a sudden unless some unwanted force is applied to the construction. For instance, a sudden gust of wind would create a force field on a stack that is equivalent to that produced by a tilted gravitational field. Such an explanation would easily apply to the findings of the second experiment. In that experiment subjects are explicitly asked if a stack of blocks looks "stable". This is an ambiguous question because the stability of a structure is always judged by imagining what would happen to the structure if an external perturbation is applied. The right question should be: "do you think this structure would fall if unperturbed". However, if stability is judged in the face of possible external perturbations then a tall structure would certainly be judged as less stable than a short structure occupying the same ground area. This is what the authors find. What they consider as a bias (tall structures are perceived as less stable than short structures) is instead a wrong interpretation of the mental process that determines stability. If subjects are asked the question "Is it going to fall?" then tall stacks of sound structure would be judged as stable as short stacks, just more precarious.

R3: Indeed, the external forces suggested by the reviewer certainly influence judgments of objects’ stability. The critical question, however, is whether humans’ judgments on objects’ stability accurately mirror the actual stability of objects in the absence of external forces. To address this question, we designed two new experiments.

Experiment 1: we duplicated the original experimental setup with the addition of a wall implemented on one side (Supplementary Figure 4A). We explicitly informed the participants that the wall could block wind, ensuring that no potential wind from the direction of the wall could influence the configuration. If participants’ judgments were affected by external noise, we would expect to observe a skewed angle distribution. Contrary to this prediction, our results showed a normal distribution across all three participants (Age: 25-30, two females), which is similar to the experiment without the wall (Supplementary Figure 4B).

**Author response image 6. sa2fig6:** Wall experiment to test the impact of external forces on the measurement of stochastic gravity. (a) Experimental setting. We replicated the original setup with the addition of a wall implemented on one side. Left: the overall experimental scene; Right, the scene shown to participants. (b) Human behaviors. Three participants conducted this experiment, and their responses consistently showed normal distributions without any skewness, suggesting that their judgments were not affected by the presence of the wall. These results support our claim that humans’ judgments on stability were not affected by potential concerns regarding external forces.

Experiment 2: The second experiment adopted another paradigm to test the hypothesis of stochastic mental simulation. Consider humans to infer the landing point of a parabolic trajectory that was obscured by an occlude (Author response image 2A), the stochastic mental simulation predicted that humans’ behavior follows a Gaussian distribution. However, if humans’ judgments were influenced by external noise, the landing points could not be Gaussian. The experiment consists of 100 trials in total, and in each trial participants used a mouse to predict the landing point of each trajectory by clicking the left button. Our results found all three participants (1 female; ages: 24-30) were unable to accurately predict the landing points of the trajectories, and the predictive errors conformed to Gaussian distributions with different variances (Author response image 2B). Therefore, this new experiment confirms the stochastic nature of intuitive physics.

**Author response image 7. sa2fig7:** Trajectory experiment to test the stochastic nature of gravity represented in the mind. (a) Experiment design. In this experiment, participants were required to use a mouse to determine the landing point of a parabolic trajectory (marked by the green dot), obscured by a grey rectangle. Note that the parabolic trajectory was determined only by gravity, and no external disturbances were introduced. The parameters used in this experiment are detailed in the upper right corner. (b) Predictive errors from three participants. The predictive errors from all three participants conform to Gaussian distributions with non-negligible variances. These results suggest the notion of an inherent stochastic property of gravity represented in the mind.

(4) The RL model used as a proof of concept for how the brain may build a stochastic prior for the direction of gravity is based on very strong and unverified assumptions. The first assumption is that the brain already knows about the force of gravity, but it lacks knowledge of the direction of this force of gravity. The second assumption is that before learning the brain knows the effect of a gravitational field on a stack of blocks. How can the brain simulate the effect of a non-vertical gravitational field on a structure if it has never observed such an event?

R4: We agree with the reviewer that the RL framework serves primarily as a theoretic model to explain the stochastic nature of the world model on gravity, rather than as a demonstration of the developmental origins of intuitive physics abilities. The genesis of such abilities is multifaceted and unlikely to be fully replicated through a simple simulation like RL. Therefore, the purpose of incorporating the RL framework in our study is to demonstrate that external perturbances are not necessary for the development of a stochastic representation of gravity.

In revision, we have clarified the role of the RL framework

Line 265-277: “While the cognitive impenetrability and the self-consistency observed in this study, without resorting to an external perturbation, favor the stochastic model over the deterministic one, the origin of this stochastic feature of the world model is unclear.

Here we used a reinforcement learning (RL) framework to unveil this origin, because our intelligence emerges and evolves under the constraints of the physical world. Therefore, the stochastic feature may emerge as a biological agent interacts with the environment, where the mismatches between external feedback from the environment and internal expectations from the world model are in turn used to fine-tune the world model (Friston et al., 2021; MacKay, 1956; Matsuo et al., 2022). Note that a key aspect of the framework is determining whether the stochastic nature of the world model on gravity emerges through this interaction, even in the absence of external noise.”

(5) The third assumption is that from the visual input, the brain is able to figure out the exact 3D coordinates of the blocks. This has been proven to be untrue in a large number of studies. Given these assumptions and the fact that the only parameters the RL model modifies through learning specify the direction of gravity, I am not surprised that the model produces the desired results.

**Author response image 8. sa2fig8:** Perception Uncertainty in 3D stacks structures. (a) Experimental design. A pair of two stacks with similar placements of blocks were presented sequentially to participants, who were instructed to judge whether the stacks were identical and to rate their confidence in this judgment. Each stack was presented on the screen for 2 seconds. (b) Behavior Performance. Three participants (2 males, age range: 24-30) were recruited to the experiment. The confidence in determining whether a pair of stacks remained unchanged rapidly decreased when each block had a very small displacement, suggesting humans could keenly perceive trivial changes in configurations. The x-axis denotes the difference in block placement between stacks, with the maximum value (0.4) corresponding to the length of a block’s short side. The Y-axis denotes humans’ confidence in reporting no change. The red curve illustrates the average confidence level across 4 runs, while the yellow curve is the confidence level of each run.

R5: Indeed, uncertainty is inevitable when perceiving the external world, because our perception is not a faithful replica of external reality. A more critical question pertains to the accuracy of our perception in representing the 3D coordinates of a stack’s blocks. To address this question, we designed a straightforward experiment (Author response image 5a), where participants were instructed to determine whether a pair of stacks were identical. The position of each block was randomly changed horizontally. We found that all participants were able to accurately identify even minor positional variations in the 3D structure of the stacks (Author response image 5b). This level of perceptual precision is adequate for locating the difference between predictions from mental simulations and actual observations of the external world.

(6) Finally, the argument that the MGS is more efficient than the NGS model is based on an incorrect analysis of the results of the simulation. It is true that 80% accuracy is reached faster by the MGS model than the 95% accuracy level is reached by the NGS model. But the question is: how fast does the NGS model reach 80% accuracy (before reaching the plateau)?

R6: Yes. The NGS model achieved 80% accuracy as rapidly as the MGS model. However, the NGS model required a significantly longer period to reach the plateau crucial for decision-making. In revision, this information is now included.

Line 348-350: “…, while the initial growth rates of both models were comparable, the MGS reached the plateau crucial for decision-making sooner than the NGS.”

We greatly appreciate the thorough and insightful review provided by all three reviewers, which has considerably improved our manuscript, especially in terms of clarity in the presentation of the approach and further validation of the robustness implications of our results.

Reference:Allen KR, Smith KA, Tenenbaum JB. 2020. Rapid trial-and-error learning with simulation supports flexible tool use and physical reasoning. Proceedings of the National Academy of Sciences 117:29302–29310.

Battaglia PW, Hamrick JB, Tenenbaum JB. 2013. Simulation as an engine of physical scene understanding. Proceedings of the National Academy of Sciences 110:18327–18332.

Friston K, Moran RJ, Nagai Y, Taniguchi T, Gomi H, Tenenbaum J. 2021. World model learning and inference. Neural Networks 144:573–590.

Kriegeskorte N, Douglas PK. 2019. Interpreting encoding and decoding models. Current opinion in neurobiology 55:167–179.

MacKay DM. 1956. The epistemological problem for automataAutomata Studies.(AM-34), Volume 34. Princeton University Press. pp. 235–252.

Matsuo Y, LeCun Y, Sahani M, Precup D, Silver D, Sugiyama M, Uchibe E, Morimoto J. 2022. Deep learning, reinforcement learning, and world models. Neural Networks.

Naselaris T, Kay KN, Nishimoto S, Gallant JL. 2011. Encoding and decoding in fMRI. Neuroimage 56:400–410.

Zhou L, Smith K, Tenenbaum J, Gerstenberg T. 2022. Mental Jenga: A counterfactual simulation model of physical support.